# GridMix: Exploring Spatial Modulation for Neural Fields in PDE Modeling

**Honghui Wang, Shiji Song & Gao Huang**[*]
Department of Automation, BNRist, Tsinghua University, Beijing, China,
`wanghh20@mails.tsinghua.edu.cn,{shijis, gaohuang}@tsinghua.edu.cn`

## Abstract

Significant advancements have been achieved in PDE modeling using neural fields. Despite their effectiveness, existing methods rely on global modulation, limiting their ability to reconstruct local details. While spatial modulation with vanilla grid-based representations offers a promising alternative, it struggles with inadequate global information modeling and overfitting to the training spatial domain. To address these challenges, we propose GridMix, a novel approach that models spatial modulation as a mixture of grid-based representations. GridMix effectively explores global structures while preserving locality for fine-grained modulation. Furthermore, we introduce spatial domain augmentation to enhance the robustness of the modulated neural fields against spatial domain variations. With all these innovations, our comprehensive approach culminates in MARBLE, a framework that significantly advancing the capabilities of neural fields in PDE modeling. The effectiveness of MARBLE is extensively validated on diverse benchmarks encompassing dynamics modeling and geometric prediction. The code will be available on https://github.com/LeapLabTHU/GridMix.

## 1 Introduction

In recent years, deep learning has attracted considerable interest in its ability to solve partial differential equations (PDEs), providing data-driven approaches to approximate solutions across a broad spectrum of problems (Lu et al., 2021; Li et al., 2021; Brandstetter et al., 2022). One particularly innovative direction within this field is the use of neural fields, or implicit neural representations (INRs) (Park et al., 2019; Sitzmann et al., 2020), which offer continuous parameterizations of PDE solutions, enabling high-resolution modeling and flexible predictions across diverse geometries.

Building upon this foundation, INR-based methods (Yin et al., 2022; Serrano et al., 2023) have demonstrated notable advancements in PDE modeling. These methods leverage INRs to encode solutions—expressed as functions of spatial coordinates—into a low-dimensional latent space via auto-decoding (Park et al., 2019). The solution operator is then approximated by learning how these latent representations evolve over time or in response to varying conditions. This framework effectively bridges the gap between complex spatial domains and the underlying PDEs, achieving impressive efficiency and accuracy in both steady-state and transient scenarios.

Despite their potential, INR-based methods face limitations in accurately representing complex spatial variations in PDE solutions, often missing fine-grained local details. This issue stems from the use of global modulation (Perez et al., 2018), a technique that allows a base INR to represent different functions by altering its behavior conditioned on function-specific latent representations. Specifically, these latent representations are utilized to create function-specific modulation parameters that modify the hidden activations of INRs, thereby altering the output. As these parameters are typically shared across all spatial positions, the model's ability to learn detailed local features and variations is hindered (Bauer et al., 2023). Additionally, INRs are susceptible to spectral bias (Rahaman et al., 2019), which can restrict their capability to learn high-frequency components of solutions.

To address the limitations of global modulation, we propose extending it to spatial modulation, drawing inspiration from the success of grid-based representations in computer vision (Liu et al.,

---
[*]Corresponding author

2020; Müller et al., 2022). Grid-based representations discretize continuous space into a regular grid, associating each grid point with a learnable vector that encodes local features. By applying this concept to PDE modeling, we can assign grid-based modulation parameters to each function, enabling a more fine-grained representation. However, while grid-based representations have proven effective in mitigating spectral bias in vision tasks under the assumption of a fixed and regular spatial domain, their straightforward adaptation to spatial modulation for PDE modeling presents new challenges. To be precise, although using independent modulation parameters at each location excels at capturing local information, it often compromises the ability to model global structures in sparse or irregular domains, resulting in poor reconstruction quality in unseen regions. This trade-off limits their generalization across varying spatial domains, which is crucial for robust PDE modeling.

In this paper, we introduce *GridMix*, a novel spatial modulation technique that builds on the strengths of grid-based representations while tackling the aforementioned challenges. We achieve this by representing spatial modulation as a mixture of basis functions, where each is parameterized through grid-based representations. GridMix preserves the fine-grained locality of grid-based modulation, allowing for detailed exploitation of local information in function reconstruction. At the same time, it regularizes the modulation space to the linear span of a small set of learned grid basis functions, which are shared across function instances and facilitate the extraction of global structural information. This regularization effectively mitigates the risk of overfitting to specific spatial domains. Additionally, we introduce *Spatial Domain Augmentation*, which simulates domain variations during training to further enhance generalization to unseen regions. Together, these innovations form the GridMix Augmented Coordinate-based Neural Fields (*MARBLE*) for PDE modeling, significantly improving the learning capacity of previous approaches that relied on global modulation.

Our contributions are summarized as follows:

1. We introduce GridMix to mitigate the spatial domain over-fitting of spatial modulation.

2. We present the MARBLE framework, which combines GridMix with spatial domain augmentation to boost the performance of existing INR-based methods for PDE modeling.

3. We demonstrate the effectiveness and versatility of MARBLE with extensive experiments, including dynamics modeling and geometric prediction.

## 2 RELATED WORK

**Neural PDE Solvers.** Deep learning has rapidly advanced as a powerful tool for solving Partial Differential Equations (PDEs) across diverse fields, including fluid dynamics (Kochkov et al., 2021; Sun et al., 2023), solid mechanics (Samaniego et al., 2020; Nguyen-Thanh et al., 2020), and inverse problems (Lu et al., 2022; Molinaro et al., 2023). Neural Operators have emerged as a leading approach within this domain, modeling the solution operator between input and output function spaces (Lu et al., 2021; Li et al., 2021). Recent research has centered on addressing key challenges for Neural PDE solvers, such as handling intricate geometries and ensuring precise predictions. To tackle these issues, methods like Geo-FNO (Li et al., 2023b) and MP-PDE (Brandstetter et al., 2022) have been developed to handle irregular grids, while Factorized-FNO (Tran et al., 2023) refines FNO's architecture and training strategy for improved performance.

**Neural Fields,** also known as implicit neural representations (INRs), utilize neural networks (typically MLPs) to represent continuous signals based on spatial coordinates. These models have demonstrated exceptional performance in various vision tasks, including image representation (Sitzmann et al., 2020; Dupont et al., 2022), 3D shape modeling (Park et al., 2019; Tancik et al., 2020), and novel view synthesis (Mildenhall et al., 2021). In the realm of PDE solving, INRs have been integrated into physics-informed neural networks (PINNs) (Raissi et al., 2019; Wang et al., 2023; 2024) to approximate solutions. While effective, traditional PINNs require training a separate model for each unique set of initial and boundary conditions. To address this limitation, recent research has explored modulating the intermediate features of a base INRs using instance-specific latent codes, enabling the representation of a solution space (Yin et al., 2022; Serrano et al., 2023). These INR-based approaches are independent to spatial discretization and scale well to high-dimensional problems. However, INRs are susceptible to spectral bias (Rahaman et al., 2019), preferentially learning low-frequency components, which can hinder their representational capacity.

**Grid-based Representations** offer a compelling solution to the aforementioned limitation of INRs (Liu et al., 2020; Takikawa et al., 2021; Müller et al., 2022; Lee et al., 2024b). These approaches discretize the input domain into a fine-grained grid, assigning a learnable feature vector to each grid cell. When querying points within this space, local feature vectors are aggregated with interpolation and integrated into INRs, either as input (Müller et al., 2022) or within intermediate layers (Lee et al., 2024b). The inherent locality of grid-based representations enables INRs to effectively capture high-frequency signals. While existing grid-based methods excel at reconstructing fine details within fixed domains, their generalization to unseen regions is restricted, hindering their applicability to PDE solving. To address this challenge, we propose learning grid features as mixtures of basis representations, promoting domain generalization while preserving locality. A related approach, Factor Fields (Chen et al., 2023), decomposes signals into components like coefficient fields and basis functions. However, while Factor Fields operates in the signal space, our work applies a similar decomposition in the modulation space. Additionally, Factor Fields focuses on tasks outside the PDE domain, whereas our approach targets PDE-related challenges, emphasizing domain generalization of spatial modulations.

## 3 METHOD

We introduce MARBLE (GirdMix Augmented Coordinate-based Neural Fields) for PDE modeling. We begin with the problem formulation in Section 3.1. Next, we provide a preliminary overview of CORAL (Serrano et al., 2023), an INR-based method for solving PDEs, in Section 3.2. Finally, we describe the spatial modulation with MARBLE and the spatial domain augmentation in Section 3.3.

### 3.1 PROBLEM DESCRIPTION

We focus on PDE modeling tasks that involve approximating the operator $\mathcal{G}^*$, which maps functions from input space $\mathcal{A} \subset L^2(\Omega, \mathbb{R}^{d_a})$ to output space $\mathcal{U} \subset L^2(\Omega, \mathbb{R}^{d_u})$ according to the governing PDE. Here $L^2(\Omega, \mathbb{R}^{d_x})$ denotes the infinite-dimensional space of square-integrable functions with domain $\Omega$ and range in $\mathbb{R}^{d_x}$. Two such exemplary tasks are investigated: 1) **Dynamics Modeling**: Here, the objective is to capture the temporal evolution of a physical system over a forecasting horizon. This translates to modeling the transition from state $u_t$ to state $u_{t+\delta t}$, where $\delta t$ represents the time step. 2) **Geometric Prediction**: This task involves making predictions based solely on the geometric configuration of the system. In the context of geometric prediction task, each data sample is observed on a unique domain $\mathcal{X}_i$. Conversely, for modeling system dynamics, a single domain $\mathcal{X}_{tr}$ is employed for training the model across all examples, while a separate domain $\mathcal{X}_{te}$ is used for evaluating the model's performance during testing. $\mathcal{X}_{tr}$ and $\mathcal{X}_{te}$ are subsets of a full domain $\mathcal{X}_{full}$.

### 3.2 PRELIMINARY

**The CORAL Framework** addresses PDE modeling tasks through a two-stage training. In the first stage, the *reconstruction* stage, CORAL uses two modulated INRs, $f_{\theta_a, \phi_a}$ and $f_{\theta_u, \phi_u}$, to parametrize input and output functions, respectively. Each of these INRs serves as a base model that is modulated with function-specific parameters to reconstruct different functions. Specifically, INR parameters $\theta_a$ and $\theta_u$ are shared across all functions within their respective space, while the modulation parameters, $\phi_{a_i}$ and $\phi_{u_i}$, are unique to each function. The modulation parameters are derived from low-dimensional latent codes $z_{a_i}$ and $z_{u_i}$ through hypernetworks $h_a$ and $h_u$: $\phi_{a_i} = h_a(z_{a_i})$ and $\phi_{u_i} = h_u(z_{u_i})$. After training, each function is represented by a compact latent code, with the reconstruction error indicating the quality of the representation. In the second stage, the *forecasting* stage, CORAL learns a mapping between latent codes $z_{a_i}$ and $z_{u_i}$ using a processing network $g_\psi$.

During inference, CORAL operates in three steps, as shown in Figure 1. First, the encoder $e_a : \mathcal{A} \mapsto \mathbb{R}^{d_z}$ projects input $a_i$ into latent space ($z_{a_i}$) via auto-decoding (Park et al., 2019), as illustrated in Figure 1(b). Given a base INR $f_{\theta_a, \phi_a}$, $e_a$ encodes the function $a_i$ by optimizing the latent code $z_{a_i}$ to minimize the reconstruction error:

$$\mathcal{L}_{\mathcal{X}_{tr}}(f_{\theta_a, \phi_{a_i}}, a_i) = \mathbb{E}_{x \sim \mathcal{X}_{tr}} \| f_{\theta_a, \phi_{a_i}}(x) - a_i(x) \|^2, \quad \text{where } \phi_{a_i} = h_a(z_{a_i}). \quad (1)$$

Next, the model $g_\psi : \mathbb{R}^{d_z} \mapsto \mathbb{R}^{d_z}$ transforms $z_{a_i}$ into the output latent code $z_{u_i}$. Finally, the decoder $\xi_u : \mathbb{R}^{d_z} \mapsto \mathcal{U}$ decodes the processed code back into the output function space through a single forward pass, as shown in Figure 1(c).

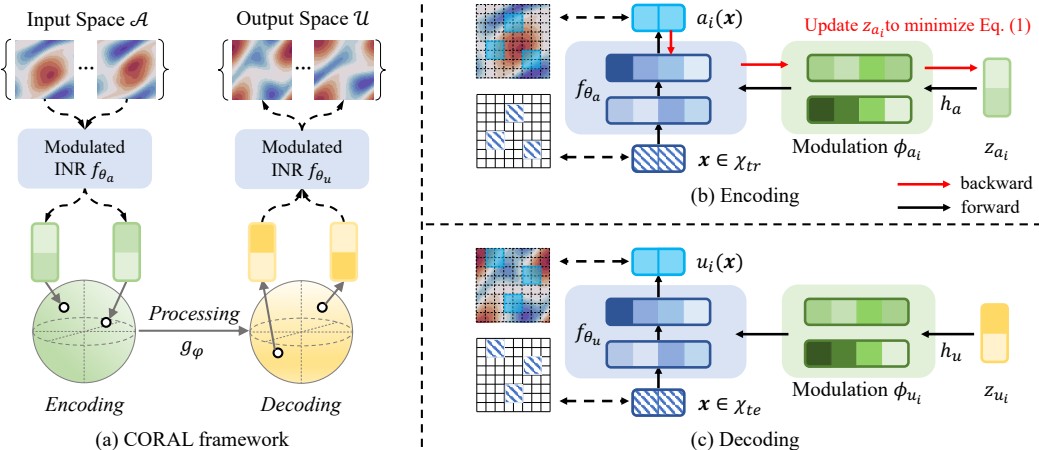

Figure 1: INR-based framework for PDE modeling.

**Meta-Learned INRs.** Typically, the encoding process could require thousands steps of gradient descent. To reduce this fitting cost, CORAL adopts the Model-Agnostic Meta Learning (MAML) algorithm (Finn et al., 2017) to meta-learn the parameters $\theta_a$ such that the encoding process can be done in a few gradient steps (Dupont et al., 2022). More specifically, a step of the MAML inner loop optimizes a latent code $z_{a_i}$ to fit a randomly sampled function $a_i$ as:

$$z_{a_i}^{(0)} = 0 \, ; z_{a_i}^{(k+1)} = z_{a_i}^{(k)} - \alpha \nabla_{z_{a_i}^{(k)}} \mathcal{L}_{\mathcal{X}_{tr}}(f_{\theta_a, \phi_{a_i}^{(k)}}, a_i), \text{ with } \phi_{a_i}^{(k)} = h_a(z_{a_i}^{(k)}) \text{ for } 0 \leq k \leq K-1, \quad (2)$$

where $\alpha$ is the inner-loop learning rate and $K$ the number of gradient steps. The outer loop then meta-learn the shared parameter $\theta_a$ by

$$\theta_a \leftarrow \theta_a - \beta \nabla_\theta \sum_{i=1}^{N} \mathcal{L}_{\mathcal{X}_{tr}}(f_{\theta_a, \phi_{a_i}^{(K)}}, a_i), \quad (3)$$

where $\beta$ denotes the outer loop learning rate and $N$ the number of samples.

**Global Modulation in CORAL.** CORAL utilizes SIRENs (Sitzmann et al., 2020) as the backbone for its INRs. SIRENs are multilayer perceptrons with sine activations, which can be formulized as

$$f_\theta(x) = \boldsymbol{W}_L\big(\sigma_{L-1} \circ \sigma_{L-2} \circ \cdots \circ \sigma_0(x)\big) + \boldsymbol{b}_L, \text{ with } \sigma_i(\eta_i(x)) = \sin\big(\omega_0(\boldsymbol{W}_i\eta_i(x) + \boldsymbol{b}_i)\big), \quad (4)$$

where $\theta = (\boldsymbol{W}_i, \boldsymbol{b}_i)_{i=0}^{L}$ are network weights and biases, $\omega_0$ is a positive scaling factor, $\eta_0(x) = x$ and $\eta_i(x)$ are activations of the $i$-th hidden layer given coordinate $x$. CORAL applys shift modulations (Perez et al., 2018) to SIRENs to represent individual function as

$$f_{\theta,\phi}(x) = \boldsymbol{W}_L(\sigma_{L-1} \circ \sigma_{L-2} \circ \cdots \circ \sigma_0(x)) + \boldsymbol{b}_L, \text{ with } \sigma_i(\eta_i(x)) = \sin\big(\omega_0(\boldsymbol{W}_i\eta_i(x) + \boldsymbol{b}_i + \phi_i)\big), \quad (5)$$

where $\phi = (\phi_i)_{i=1}^{L-1}$ represents modulation parameters at the $i$-th layer. Note that the global modulations $\phi_i$ are shared across different spatial coordinates as shown in Figure 2(a). This property limits the capacity of modulated INRs to represent complex function space. As demonstrated in previous work (Bauer et al., 2023), this global modulation fails to capture local details because any changes in modulations will lead to global perturbations across the reconstructed function.

### 3.3 SPATIAL MODULATION WITH GRIDMIX

While the original modulation design only incorporates global information of individual function to adjust the behavior of the based INR, we propose an enhanced approach which leverages local information for fine-grained modulation, without compromising its ability to utilize global information.

**Spatial Modulation Framework.** Building on prior work (Müller et al., 2022; Lee et al., 2024b) that highlights the effectiveness of grid-based representations in enhance the learning capacity of INRs, we extend the global modulation scheme to a spatial modulation approach (Figure 2(b)). We achieve this by introducing a single-channel grid-based representation $\phi_i \in \mathbb{R}^{H \times W}$, where $H$ and

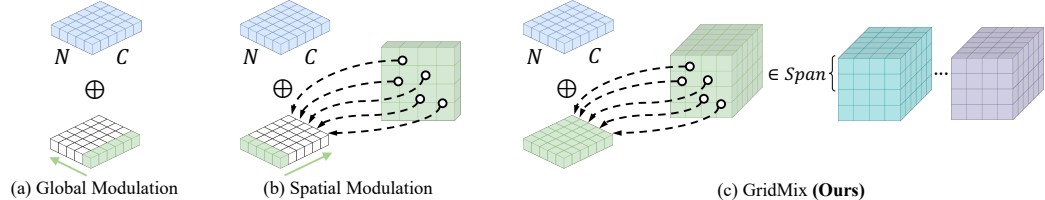

(a) Global Modulation     (b) Spatial Modulation     (c) GridMix **(Ours)**

Figure 2: Different modulation strategies. We visualize the hidden activations and modulation parameters as blue and green tensors, respectively. The hidden activations has dimensions $N \times C$, where $N$ represents the number of query locations and $C$ the number of channels. The green arrow indicates the dimension along which the modulation parameters are broadcast.

$W$ are spatial resolution of the grid (considering a 2D domain as an example). To derive the position-dependent shift modulation scalar $\phi_i(x)$ for a given spatial location $x$ from this grid, we first identify the neighboring grid points surrounding this location (in 2D, this involves four points). We then compute $\phi_i(x)$ by interpolating the values at these neighboring points, using bilinear interpolation in the 2D case. The spatial modulation reformulates Equation 5 as

$$\sigma_i(\eta_i(x)) = \sin\big(\omega_0(\boldsymbol{W}_i\eta_i(x) + \boldsymbol{b}_i + \phi_i(x))\big). \tag{6}$$

By assigning learnable parameters to each grid cell, this approach enables the model to effectively capture and adapt to spatially varying features.

While the localized nature of spatial modulation enhances learning capacity, it also introduces limitations, particularly in capturing global information. The flexibility of grid-based modulation allows meta-learned INRs to minimize the reconstruction error $\mathcal{L}_{\mathcal{X}_{tr}}(f_{\theta_a, \phi_{a_i}}, a_i)$ independently at each location $x \in \mathcal{X}_{tr}$. However, this localized focus can overlook broader contextual relationships essential for a comprehensive understanding of the data. As a result, the model may become overly specialized to the training domain $\mathcal{X}_{tr}$, leading to over-fitting. This over-fitting impairs the model's ability to generalize effectively to new data in unseen test domains $\mathcal{X}_{te}$.

To further illustrate this limitation, we compare the performance of INRs with global and local modulations under two settings: (1) the test function is observed on the training domain $\mathcal{X}_{tr}$ and reconstructed on $\mathcal{X}_{full}$; and (2) the function is observed on test domain $\mathcal{X}_{te}$. The results in Table 1 and Figure 3 clearly demonstrate that, although spatial modulation significantly reduces reconstruction error on $\mathcal{X}_{tr}$, it fails to capture the global information of test data, resulting in a large error on $\mathcal{X}_{full}$. Moreover, it performs poorly

Table 1: **Comparing different modulations** on *Navier-Stokes* with irregular grid ($\pi = 20\%$). We use the data on observed domain to optimize the modulation parameters via Eq. (2) and evaluate the reconstruction error on inference domain with the optimized modulations. Metrics in MSE.

| Observed Domain → | $\mathcal{X}_{tr}$ | | $\mathcal{X}_{te}$ | |
|---|---|---|---|---|
| Inference Domain → | $\mathcal{X}_{tr}$ | $\mathcal{X}_{full}$ | $\mathcal{X}_{te}$ | $\mathcal{X}_{full}$ |
| Global Modulation | 1.32e-4 | 1.45e-4 | 3.48e-3 | 3.72e-3 |
| Spatial Modulation | 3.95e-5 | 5.78e-2 | 4.08e-1 | 4.67e-1 |
| Single-channel GridMix | 3.17e-3 | 3.88e-3 | 9.56e-3 | 9.96e-3 |
| Multi-channel GridMix | 1.90e-5 | 2.85e-5 | 1.94e-3 | 2.25e-3 |
| Global Modulation + *SDA* | 2.88e-4 | 3.15e-4 | 4.50e-4 | 4.85e-4 |
| Multi-channel GridMix + *SDA* | 7.45e-5 | 9.64e-5 | 1.38e-4 | 1.49e-4 |

with data observed on unseen domain in $\mathcal{X}_{te}$. While this limitation is less critical in vision tasks, where a fixed and regular spatial domain is typically assumed, it becomes a significant concern in the context of PDE modeling, where generalization across varying spatial domains is a crucial metric.

**GridMix.** To mitigate over-fitting associated with spatial modulation with respect to the training spatial domain, we introduce a mixture of grid-based representations for generating spatial modulations, as illustrated in Figure 2(c). We begin by defining a set of grid-based representations to serve as basis functions and construct the spatial modulations as a linear span of these basis functions. Specifically, the grid mixture in each hidden layer is given by:

$$\phi_i(x) = \sum_{m=1}^{M} c_i^m \Phi_i^m(x), \quad \text{where} \quad [\boldsymbol{c}_1, \boldsymbol{c}_2, \ldots, \boldsymbol{c}_{L-1}] = h(z) = \boldsymbol{W}_h z + \boldsymbol{b}_h. \tag{7}$$

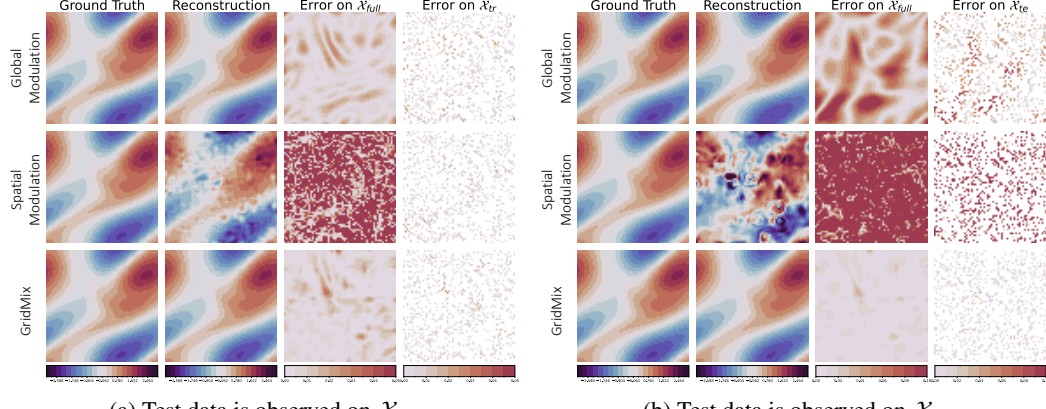

(a) Test data is observed on $\mathcal{X}_{tr}$.  (b) Test data is observed on $\mathcal{X}_{te}$

Figure 3: Comparison between INRs with global modulation (*Top*), spatial modulation with vanilla grid-based representation (*Middle*) and GridMix (*Bottom*). Each model is trained with data observed on $\mathcal{X}_{tr}$. During inference, test data is observed on (a) $\mathcal{X}_{tr}$ or (b) $\mathcal{X}_{te}$ and reconstructed on $\mathcal{X}_{full}$.

Here $(\Phi_i^m)_{m=1}^M$ denote $M$ grid basis functions, and $\boldsymbol{c}_i = (c_i^m)_{m=1}^M$ represent the mixture coefficients at the $i$-th layer. These mixture coefficients are efficiently determined from a low-dimensional embedding $z$ via a linear hypernetwork $h$. The grid basis functions are shared across different function instances and are learned alongside the modulated INRs.

This approach not only maintains the localized nature of grid-based modulation but also enables the capture of global information through the shared basis functions. The rationale behind using grid mixtures parallels the principles of spectral methods, where a problem is decomposed into simpler components that can efficiently capture both local and global features. Additionally, GridMix act as a form of regularization to the modulation space. For instance, in vanilla spatial modulation, each function is assigned $H \times W$ parameters, resulting in a high-dimensional parameter space. In contrast, GridMix reduces this dimensionality to $M$, where $M$ represents the number of shared basis functions used across all functions. This strategy effectively prevents the modulated INRs from over-fitting to the data distribution on a specific spatial domain, thereby improving the model's ability to generalize across a diverse range of previously unseen domains (see Table 1 and Figure 3).

**Channel-wise Modulations.** While GridMix effectively regularize the modulation space to prevent over-fitting, they can also limit the model's representational capacity. To address this issue, we propose incorporating channel-wise modulations into GridMix. Although a single-channel approach often performs adequately with vanilla spatial modulation, it falls short within the more constrained framework of GridMix. Therefore, we extend single-channel grid basis functions $\Phi_i^m \in \mathbb{R}^{H \times W}$ to multi-channel ones $\Phi_i^m \in \mathbb{R}^{H \times W \times C}$, where $C$ is the number of channels. This extension allows for a spatially dependent modulation vector $\phi_i(x)$, which combines the benefits of GridMix with enhanced expressiveness. Despite this increase in representational capability, the compactness of the modulation space is preserved by sharing the mixture coefficients across all channels.

**Spatial Domain Augmentation.** While our previous design improves generalization across spatial domains through modulation strategies, we now tackle the problem from a different perspective by refining the learning process itself. One inherent reason for the limited generalization is the lack of domain variation during training, as the training domain $\mathcal{X}_{tr}$ remains fixed in Equations 2 and 3. To counter this, we propose data augmentation of spatial domains to simulate domain variations encountered during inference. To illustrate, for each run of inner and outer loops, we randomly sample a subset of training coordinates $\mathcal{X}_{sub}$ from the fixed domain $\mathcal{X}_{tr}$, as described below:

$$\text{Inner loop: } z_{a_i}^{(k+1)} = z_{a_i}^{(k)} - \alpha \nabla_{z_{a_i}^{(k)}} \mathcal{L}_{\mathcal{X}_{sub} \sim \mathcal{X}_{tr}} (f_{\theta_a, \phi_{a_i}^{(k)}}, a_i), \ 0 \le k \le K-1; \tag{8}$$

$$\text{Outer loop: } \theta_a \leftarrow \theta_a - \beta \nabla_\theta \sum_{i=1}^N \mathcal{L}_{\mathcal{X}_{sub} \sim \mathcal{X}_{tr}} (f_{\theta_a, \phi_{a_i}^{(K)}}, a_i). \tag{9}$$

By incorporating spatial domain augmentation, we enhance the robustness of modulated INRs with respect to the variation of spatial domain.

## 4 EXPERIMENTS

We assessed the flexibility of our model by applying it to two different tasks: forecasting the temporal dynamics of a physical system (see Section 4.1) and predicting the steady-state behavior based on domain geometry (see Section 4.2). Both tasks were conducted using the experimental setup described in the CORAL paper (Serrano et al., 2023), with additional details available in Appendix B. Moreover, a thorough ablation study of our method is presented in Section 4.3.

### 4.1 DYNAMICS MODELING

The objective of dynamics modeling is to forecast the temporal evolution of a physical system over a specified horizon. To accomplish this, we follow an autoregressive framework operating within a latent space. Formally, given a sequence of target functions $(u_0, u_{\delta t}, ..., u_{T\delta t})$, where $u_t$ represents the system state at time $t$ and $\delta t$ the time step, our aim is to predict the subsequent states $u_{k\delta t}$ for $k = 1, ..., T$ based on the initial condition $u_0$. This is achieved by encoding $u_0$ into a latent representation $z_0 = e(u_0)$ and recursively generating subsequent latent codes $z_{k\delta t}$ using an autoregressive processor $g_\psi$. Finally, the predicted latent states are decoded to obtain the corresponding estimated functions $\hat{u}_{k\delta t} = \xi(z_{k\delta t})$, $k = 1, ..., T$.

**Architectures.** For this problem, we employ a single modulated INR $f_{\theta,\phi}$ to represent the function space encompassing the physical quantity across diverse initial conditions and time instances. The modulated INR is trained by the meta-learning algorithm detailed in Section 3. For the autoregressive processor $g_\psi$, a Neural Ordinary Differential Equation (NODE) (Chen et al., 2018) is adopted, enabling flexible predictions at arbitrary time steps. Given an initial latent state $z_t$, the NODE solver recursively computes the latent code at desired time steps $t + \tau$ according to the equation $z_{t+\tau} = g_\psi(z_t, \tau) = z_t + \int_t^{t+\tau} \zeta_\psi(z_s)ds$, where $\zeta_\psi$ is a neural network parameterized by $\psi$. The integral term is approximated using a fourth-order Runge-Kutta scheme. Training of the NODE is formulated as the minimization problem:

$$\arg\min_\psi \mathbb{E}_{u\sim\nu_u, t\sim\mathcal{U}(0,T]}\|g_\psi(e(u_0), t) - e(u_t)\|^2, \tag{10}$$

where $\nu_u$ denotes the distribution of physical quantity trajectories.

**Datasets and Baselines.** We generate two datasets of fluid dynamics using the 2D Navier-Stokes equation (*Navier-Stokes*), which models a viscous, incompressible fluid on a regular domain, and the 3D spherical Shallow-Water equation (*Shallow-Water*), which describes the movements of the Earth's atmosphere. These datasets consist of time-evolving vorticity fields (and height for *Shallow-Water*), derived from diverse initial conditions. More details can be found in Appendix A.1. Several state-of-the-art models were compared, encompassing two neural operator methods (DeepONet (Lu et al., 2021) and FNO (Li et al., 2021)), a mesh-based network (MP-PDE (Brandstetter et al., 2022)), and two coordinate-based approaches (DINo (Yin et al., 2022) and CORAL (Serrano et al., 2023)). The baseline results for comparison are sourced from Serrano et al. (2023).

**Evaluation Criteria.** To assess the model's spatiotemporal generalization capabilities, two evaluation protocols are employed. ● **Temporal Extrapolation**: Each tracjectory is partitioned into two equal-length sub-trajectories of 20 timestamps: an in-time (*In-t*) and out-of-time (*Out-t*) segment. The model is trained to forecast up to the end of the *In-t* segment and evaluated on both horizons. This assesses the model's ability to predict within and beyond the training regime. ● **Spatial Subsampling**: From the original domain $\mathcal{X}_{full}$, with a resolution of 64×64 for *Navier-Stokes* and 64×128 for *Shallow-Water*, we randomly select $\pi$ percent of points as the training grid $\mathcal{X}_{tr}$. A distinct grid $\mathcal{X}_{te}$ with identical sparsity is created for testing. Both grids remain constant across different trajectories, allowing for evaluation of the model's ability to generalize to unseen locations.

**Results.** Table 2 comprehensively evaluate the temporal extrapolation capabilities of various machine learning models on the *Navier-Stokes* and *Shallow-Water* datasets under different grid densities (100%, 20%, and 5%). Our method consistently yields the best or near-best Mean Squared Error,

Table 2: **Dynamics Modeling** - Test results. Metrics in MSE. The best results are in **bold**, while the second-best results are underlined. The "lin. int." abbreviates linear interpolation of irregular grid data onto a regular grid. The term "n.a." indicates "not available", as the *Shallow-Water* data is in a 2D spherical grid (latitude and longitude), making it incompatible with linear interpolation.

| $\mathcal{X}_{tr} \downarrow \mathcal{X}_{te}$ | dataset → | *Navier-Stokes* | | *Shallow-Water* | |
|---|---|---|---|---|---|
| | | *In-t* | *Out-t* | *In-t* | *Out-t* |
| | DeepONet | 4.72e-2 ± 2.84e-2 | 9.58e-2 ± 1.83e-2 | 6.54e-3 ± 4.94e-4 | 8.93e-3 ± 9.42e-5 |
| | FNO | 5.68e-4 ± 7.62e-5 | 8.95e-3 ± 1.50e-3 | 3.20e-5 ± 2.51e-5 | **1.17e-4 ± 3.01e-5** |
| $\pi = 100\%$ | MP-PDE | 4.39e-4 ± 8.78e-5 | 4.46e-3 ± 1.28e-3 | 9.37e-5 ± 5.56e-6 | 1.53e-3 ± 2.62e-4 |
| regular grid | DINo | 1.27e-3 ± 2.22e-5 | 1.11e-2 ± 2.28e-3 | 4.48e-5 ± 2.74e-6 | 2.63e-3 ± 1.36e-4 |
| | CORAL | 1.86e-4 ± 1.44e-5 | 1.02e-3 ± 8.62e-5 | 3.44e-6 ± 4.01e-7 | 4.82e-4 ± 5.16e-5 |
| | MARBLE (Ours) | **3.52e-5 ± 5.31e-6** | **5.04e-4 ± 3.68e-5** | **8.21e-7 ± 1.13e-8** | 1.42e-4 ± 7.07e-6 |
| | DeepONet | 8.37e-1 ± 2.07e-2 | 7.80e-1 ± 2.36e-2 | 1.05e-2 ± 5.01e-4 | 1.09e-2 ± 6.16e-4 |
| | FNO + lin. int. | 3.97e-3 ± 8.03e-4 | 9.92e-3 ± 2.36e-3 | n.a. | n.a. |
| $\pi = 20\%$ | MP-PDE | 3,98e-2 ± 1,69e-2 | 1,31e-1 ± 5,34e-2 | 5.28e-3 ± 5.25e-4 | 2.56e-2 ± 8.23e-3 |
| irregular grid | DINo | 9.99e-4 ± 6.71e-3 | 8.27e-3 ± 5.61e-3 | 2.20e-3 ± 1.06e-4 | 4.94e-3 ± 1.92e-4 |
| | CORAL | 2.18e-3 ± 6.88e-4 | 6.67e-3 ± 2.01e-4 | 1.41e-3 ± 1.39e-4 | 2.11e-3 ± 5.58e-5 |
| | MARBLE (Ours) | **1.62e-4 ± 2.42e-5** | **9.27e-4 ± 1.44e-4** | **7.06e-4 ± 4.60e-5** | **8.45e-4 ± 3.01e-5** |
| | DeepONet | 7.86e-1 ± 5.48e-2 | 7.48e-1 ± 2.76e-2 | 1.11e-2 ± 6.94e-4 | 1.12e-2 ± 7.79e-4 |
| | FNO + lin. int. | 3.87e-2 ± 1.44e-2 | 5.19e-2 ± 1.10e-2 | n.a. | n.a. |
| $\pi = 5\%$ | MP-PDE | 1.92e-1 ± 9.27e-2 | 4.73e-1 ± 2.17e-1 | 1.10e-2 ± 4.23e-3 | 4.94e-2 ± 2.36e-2 |
| irregular grid | DINo | 8.65e-2 ± 1.16e-2 | 9.36e-2 ± 9.34e-3 | **1.22e-3 ± 2.05e-4** | 1.52e-2 ± 3.74e-4 |
| | CORAL | 2.44e-2 ± 1.96e-2 | 4.57e-2 ± 1.78e-2 | 8.77e-3 ± 7.20e-4 | 1.29e-2 ± 1.92e-3 |
| | MARBLE (Ours) | **1.43e-3 ± 5.66e-4** | **4.73e-3 ± 1.33e-3** | 5.92e-3 ± 3.32e-4 | **5.98e-3 ± 3.51e-4** |

substantially surpassing baseline performance in most scenarios. Specifically, at 100% grid density, our method underscores the efficacy of spatial modulation, achieving up to a 5.3x reduction in error for in-time predictions and a 3.4x reduction for out-of-time predictions compared to the global modulation baseline, CORAL. Additionally, our method generally outperforms FNO in most settings and matches its performance in the *Shallow-Water Out-t* scenario. Even under the more challenging 20% and 5% grid density settings, our approach retains its superiority, showcasing the effectiveness of GridMix and spatial domain augmentation in mitigating over-fitting to training domain. Notably, on the *Navier-Stokes* dataset, it delivers robust performance with up to a 94.1% reduction in *In-t* MSE and a 89.5% reduction in *Out-t* MSE compared to the best-performing baseline. These results highlight the strength of our method in handling dynamics modeling tasks, particularly under sparse and irregular grid, where it demonstrates both accuracy and resilience. Note that DINo outperforms our method in the *In-t* case of *Shallow-Water* ($\pi = 5\%$) but underperforms in *Out-t*. We attribute this to DINo's overly-optimized latent embeddings, which lead to over-fitting on training data.

## 4.2 GEOMETRY-AWARE INFERENCE

This section explores the inference of steady-state system behavior based on domain geometry. For a system with domain $\Omega_i$, its geometry is discretely represented as a point cloud or structured mesh $\mathcal{X}_i \subset \Omega_i$. This mesh is a deformation of a reference grid $\mathcal{X}$ to fit the specific object shape, such as an airfoil. The problem is formulated as an operator learning task, where the input is the grid deformation defined on the reference grid $\mathcal{X}$ and the output is the corresponding physical quantity $u_i$ defined on $\Omega_i$. We aim to develop a model that can generalize to unseen geometries.

**Architectures.** We utilize two modulated INRs to encode the grid transformations and physical quantities, respectively. A simple MLP is employed as the processor $g_\psi$ to learn the mapping between the latent codes of these two function spaces.

**Datasets and Baselines.** To evaluate the proposed method, three benchmark datasets introduced by Li et al. (2023a) are considered, including the Euler equation (*NACA-Euler*), Navier-Stokes equation (*Pipe*) and Hyper-elastic material (*Elasticity*). More details can be found in Appendix A.2. Our method is compared against three state-of-the-art models, Geo-FNO (Li et al., 2023b), Factorized-FNO (Tran et al., 2023) and CORAL (Serrano et al., 2023), as well as two regular-grid baselines, FNO (Li et al., 2021) and UNet (Ronneberger et al., 2015), which are applied to the data after interpolation. The baseline results for comparison are sourced from Serrano et al. (2023).

Table 3: **Geometry-aware inference** - Test results. Metrics in relative L2 error. The best results are shown in **bold**, while the second-best results are underlined.

| Model | *NACA-Euler* | *Elasticity* | *Pipe* |
|---|---|---|---|
| FNO | 3.85e-2 ± 3.15e-3 | 4.95e-2 ± 1.21e-3 | 1.53e-2 ± 8.19e-3 |
| UNet | 5.05e-2 ± 1.25e-3 | 5.34e-2 ± 2.89e-4 | 2.98e-2 ± 1.08e-2 |
| Geo-FNO | 1.58e-2 ± 1.77e-3 | 3.41e-2 ± 1.93e-2 | **6.59e-3 ± 4.67e-4** |
| Factorized-FNO | 6.20e-3 ± 3.00e-4 | 1.96e-2 ± 2.00e-2 | 7.33e-3 ± 4.66e-4 |
| CORAL | 5.90e-3 ± 1.00e-4 | 1.67e-2 ± 4.18e-4 | 1.20e-2 ± 8.74e-4 |
| MARBLE (Ours) | **5.79e-3 ± 9.67e-5** | **1.12e-2 ± 9.43e-5** | 1.03e-2 ± 2.62e-4 |

**Results.** We present our results in Table 3. Notably, our method demonstrates consistent enhancement over the baseline method with global modulation across various geometry-aware inference tasks. For instance, MARBLE achieves the lowest error rate of 1.12e-2 on *Elasticity*, outperforming CORAL by 32.9%. Although it slightly underperforms compared to Geo-FNO and Factorized-FNO on *Pipe*, MARBLE still delivers an improvement of 14.2% over CORAL. These results highlight the effectiveness of GridMix in enhancing generalization to unseen geometries.

## 4.3 Ablation Study and Analysis

**Effects of Spatial Domain Augmentation and GridMix.** By incrementally incorporating the proposed mechanisms into the baseline model, CORAL, we observed a consistent improvement in performance, as illustrated in Table 4a. The spatial domain augmentation notably enhances baseline performance on unseen domains $\mathcal{X}_{te}$, demonstrating its effectiveness in mitigating over-fitting to the training domain. This approach also proves robust across varying sampling ratios (Table 4b). Building on this, the multi-channel GridMix further elevate the performance, with improvements of 60.7% and 51.0% for *In-t* and *Out-t*, respectively. These gains align with the regular grid setting, where error rates are reduced by 80.8% and 50.6%.

Table 4: **MARBLE ablation experiments** on *Navier-Stokes*. We use the irregular grid setting with $\pi = 20\%$ to assess (a) the effects of core components, including spatial domain augmentation (*SDA*) and multi-channel GridMix (*MCGM*). We also examine (b) the impact of *SDA*'s sampling ratio in this setting. Conversely, the regular grid setting is employed to analyze (c,d) GridMix design and (e) latent dimensions. The reported metric is MSE. Default settings are marked in gray.

(a) **Core components**.

| SDA | MCGM | *In-t* | *Out-t* |
|---|---|---|---|
| ✗ | ✗ | 2.18e-3 | 6.67e-3 |
| ✓ | ✗ | 4.22e-4 | 1.89e-3 |
| ✓ | ✓ | **1.62e-4** | **9.27e-4** |

(b) **Sampling ratio**.

| ratio | *In-t* | *Out-t* |
|---|---|---|
| 0.2 | 5.94e-4 | 3.40e-3 |
| 0.4 | **1.62e-4** | **9.27e-4** |
| 0.6 | 1.67e-4 | 1.27e-3 |

(c) **Grid resolution**.

| res | *In-t* | *Out-t* |
|---|---|---|
| 4 | 4.60e-5 | 6.10e-4 |
| 8 | 3.52e-5 | **5.04e-4** |
| 16 | **3.41e-5** | 5.78e-4 |
| 32 | 4.10e-5 | 1.68e-3 |

(d) **Grid basis functions**.

| num | *In-t* | *Out-t* |
|---|---|---|
| 8 | 4.02e-5 | 8.33e-4 |
| 16 | 3.93e-5 | 5.36e-4 |
| 32 | **3.52e-5** | **5.04e-4** |
| 64 | 3.84e-5 | 5.54e-4 |

(e) **Latent dimension**.

| dim | *In-t* | *Out-t* |
|---|---|---|
| 16 | 4.04e-5 | 5.49e-4 |
| 32 | **3.52e-5** | **5.04e-4** |
| 64 | 4.70e-5 | 6.19e-4 |
| 128 | 6.66e-5 | 8.81e-4 |

**GridMix Design.** Table 4c and 4d explore the impact of grid resolution and the number of grid basis functions. Our findings indicate that the performance plateaus as resolution and basis numbers increase. We hypothesize that this saturation occurs because, although higher resolution and more basis functions offer greater flexibility in encoding the function space into the latent space, they also lead to increasingly complex latent trajectories that are more challenge to learn.

**Latent Dimension.** We investigate the influence of latent code dimension in Table 4e. GridMix constrains the modulation's degrees of freedom to the number of mixture coefficients, enabling the use of a smaller latent space dimension. As demonstrated in Table 4e, the optimal performance is achieved with a latent dimension of 32. This reduced dimension results in a more compact latent space and simpler latent trajectories for dynamics modeling.

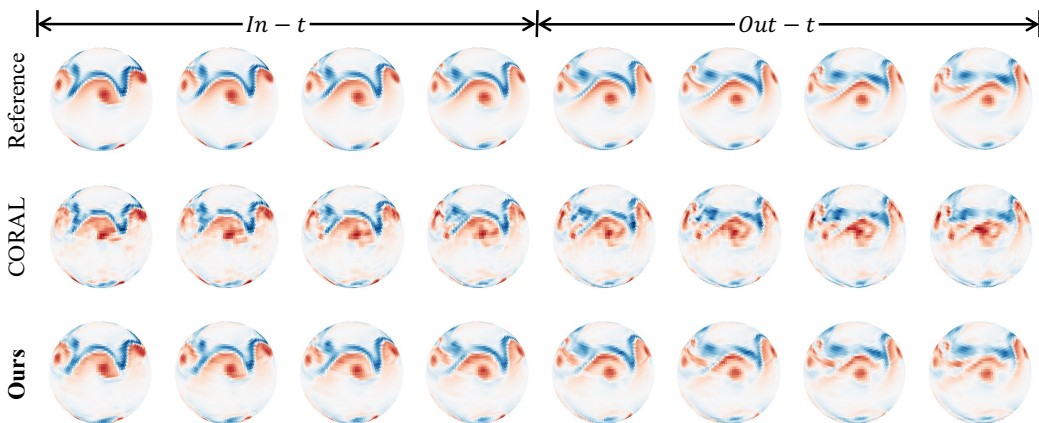

Figure 4: Visualization of predictions of MARBLE on *Shallow-Water* dataset with $\pi = 20\%$.

**Comparison with Scaled Baselines.** To account for the additional learnable parameters introduced by grid mixtures, we compare MARBLE against scaled baselines. Specifically, we evaluate CORAL with a hidden dimension of 512 (CORAL-512), which has a comparable number of parameters to MARBLE, and CORAL with a hidden dimension of 768 (CORAL-768), which has approximately twice the number of parameters as MARBLE. As

Table 5: **Comparison with scaled baselines** - Experiments are conducted on *Navier-Stokes* ($\pi = 20\%$) with spatial domain adaptation (*SDA*). *RecErr* denotes the test error on the reconstruction stage. Metrics in MSE.

| Model | # Params | *RecErr* | *In-t* | *Out-t* |
|---|---|---|---|---|
| CORAL-128 + *SDA* | 83.1K | 4.50e-4 | 4.22e-4 | 1.89e-3 |
| CORAL-512 + *SDA* | 725.5K | 1.69e-4 | 3.79e-4 | 1.95e-3 |
| CORAL-768 + *SDA* | 1,481.5K | 1.44e-4 | 1.17e-3 | 5.83e-3 |
| MARBLE (Ours) | 823.1K | **1.38e-4** | **1.62e-4** | **9.27e-4** |

shown in Table 5, the increase in parameters reduces the reconstruction error, leading to a better fit of the data space. However, this does not necessarily translate to improved dynamics forecast performance. This finding aligns with earlier research by Serrano et al. (2023), as outlined in their Table 11. These results highlight the effectiveness of MARBLE in leveraging more parameters to enhance the performance of operator learning within the INR-based framework.

**Visualizations.** Figure 4 shows the predictions of MARBLE on the irregular grid setting with $\pi = 20\%$ for the *Shallow-Water* dataset. We compare these results with CORAL, the best-performing baseline. Notably, MARBLE excels at modeling complex spatial variations in dynamics forecasting, particularly in the *Out-t* scenario. Additional visualizations can be found in Appendix C.

## 5 DISCUSSION AND CONCLUSION

This paper has presented MARBLE, a novel approach for PDE modeling that leverages the power of implicit neural representations with innovative spatial modulation techniques. Our method addresses the limitations of existing INR-based methods by introducing GridMix, a spatial modulation technique that balances the exploitation of global and local information. Additionally, we propose spatial domain augmentation to enhance generalization across varying spatial regions. Through extensive experiments, we have demonstrated the superior performance of MARBLE in various PDE modeling tasks, including dynamics modeling and geometric prediction.

While MARBLE significantly enhances learning capacity, it's important to note that the complexity of GridMix may introduce additional computational overhead and memory requirements. A promising avenue for future research would be to explore efficient implementations of GridMix, potentially drawing inspiration from vector quantization and tensor factorization techniques commonly employed in grid-based representations (Takikawa et al., 2022; Chen et al., 2022). We also observe that MARBLE performs below GEO-FNO on the *Pipe* dataset, likely due to the limitations of the SIREN backbone, as noted by Serrano et al. (2023). While SIREN is effective for isotropic frequency distributions, it may struggle with data featuring strong directional anisotropy due to its sensitivity to frequency-related hyperparameters. Future work could explore enhanced INR designs (Cho et al., 2024) that better capture directional frequency variations.

ACKNOWLEDGMENTS

The work is supported in part by the National Natural Science Foundation of China under Grant 62276150 and the Guoqiang Institute of Tsinghua University.

REPRODUCIBILITY STATEMENT

We provide detailed descriptions of dataset generation for the benchmarks in Appendix A. The experimental setups for our method are outlined in Appendix B. Additionally, our code is included in the supplementary materials.

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

## APPENDIX

## A   BENCHMARKS

### A.1   DYNAMICS MODELING

**2D-Navier-Stokes (*Navier-Stokes*).**   We examine the 2D Navier-Stokes equation as detailed in Li et al. (2021); Yin et al. (2022); Serrano et al. (2023), which simulates the behavior of an incompressible fluid within a rectangular domain $\Omega = [-1, 1]^2$. The governing PDE is given by:

$$\frac{\partial w(x,t)}{\partial t} = -u(x,t)\nabla w(x,t) + \nu\Delta w(x,t) + f, x \in [-1,1]^2, t \in [0,T] \tag{11}$$

$$w(x,t) = \nabla \times u(x,t), x \in [-1,1]^2, t \in [0,T] \tag{12}$$

$$\nabla u(x,t) = 0, x \in [-1,1]^2, t \in [0,T] \tag{13}$$

where $u$ denotes the velocity field, $w$ represents the vorticity, $\nu$ is the fluid viscosity, and $f$ is the forcing term defined as:

$$f(x_1, x_2) = 0.1\left(\sin(2\pi(x_1 + x_2)) + \cos(2\pi(x_1 + x_2))\right), \forall x \in \Omega \tag{14}$$

The system is subject to periodic boundary conditions. We generated data by sampling initial conditions as in Li et al. (2021), producing various trajectories on a $256 \times 256$ spatial grid with a time step $\delta t = 1$. To capture significant dynamics, we retain the trajectory starting from the 20th timestep. Each resulting trajectory contains 40 snapshots. We split these trajectories into two segments: the first 20 frames are utilized for training (denoted as *In-t*), while the remaining 20 frames are reserved for testing the model's extrapolation capability (referred to as *Out-t*). In total, we generated 256 trajectories for training and 16 for evaluation.

**3D-Spherical Shallow-Water (*Shallow-Water*).**   We investigate the shallow-water equation on a spherical surface, which models atmospheric dynamics on Earth:

$$\frac{du}{dt} = -f \cdot k \times u - g\nabla h + \nu\Delta u \tag{15}$$

$$\frac{dh}{dt} = -h\nabla \cdot u + \nu\Delta h \tag{16}$$

Here, $k$ is a unit vector perpendicular to the sphere's surface, $u$ is the velocity field tangent to the sphere, which can be converted to vorticity $w = \nabla \times u$, and $h$ is the sphere's height. Data generation was carried out using the *Dedalus* framework (Burns et al., 2020), following the configuration described in Yin et al. (2022), where symmetric behavior is observed in both hemispheres. The initial velocity $u_0$ consists of two symmetric bands, one in each hemisphere, aligned with the latitude circles. For each point at latitude $\phi$ and longitude $\theta \in [-\frac{\pi}{2}, \frac{\pi}{2}] \times [-\pi, \pi]$:

$$u_0(\phi, \theta) = \begin{cases} \left(\frac{u_{max}}{e_n}\exp\left(\frac{1}{(\phi-\phi_0)(\phi-\phi_1)}\right), 0\right) & \text{if } \phi \in (\phi_0, \phi_1), \\ \left(\frac{u_{max}}{e_n}\exp\left(\frac{1}{(\phi+\phi_0)(\phi+\phi_1)}\right), 0\right) & \text{if } \phi \in (-\phi_1, -\phi_0), \\ (0, 0) & \text{otherwise.} \end{cases} \tag{17}$$

where $u_{max}$ is the peak velocity, $\phi_0 = \frac{\pi}{7}$, $\phi_1 = \frac{\pi}{2} - \phi_0$, and $e_n = \exp(-\frac{4}{(\phi_1-\phi_0)^2})$. The initial water height $h_0$ is computed by solving a boundary value problem as described in Galewsky et al. (2004), with perturbation added by $h_0'$:

$$h_0'(\phi, \theta) = \hat{h}\cos(\phi)\exp\left(-\left(\frac{\theta}{\alpha}\right)^2\right)\left[\exp\left(-\left(\frac{\phi_2-\phi}{\beta}\right)^2\right) + \exp\left(-\left(\frac{\phi_2+\phi}{\beta}\right)^2\right)\right]. \tag{18}$$

where $\phi_2 = \frac{\pi}{4}$, $\hat{h} = 120$ m, $\alpha = \frac{1}{3}$, and $\beta = \frac{1}{15}$ following Galewsky et al. (2004). Simulations were executed on a latitude-longitude grid using Dedalus (Burns et al., 2020), starting with an initial grid size of $128 \times 256$, which was downsampled to $64 \times 128$. Data generation involved sampling $u_{max}$ from a uniform distribution $\mathcal{U}(60, 80)$, with snapshots taken every hour over a period of 320 hours, yielding 320 timestamps per trajectory. We created 16 trajectories for training and 2 for testing. However, since the early snapshots exhibited less dynamic activity, only the last 160 snapshots were retained. These long trajectories were then segmented into sub-trajectories of 40 timestamps each, resulting in 64 training trajectories and 8 testing trajectories. Finally, the data was rescaled: height $h$ was scaled by $3 \times 10^3$, and vorticity $w$ was scaled by a factor of 2.

## A.2 GEOMETRIC-AWARE INFERENCE

We utilize datasets from Li et al. (2023b) and use the original authors' train/test split.

**Euler's Equation (*Naca-Euler*).** This experiment focuses on transonic flow over an airfoil, governed by the Euler equations:

$$\frac{\partial \rho_f}{\partial t} + \nabla \cdot (\rho_f u) = 0, \frac{\partial \rho_f u}{\partial t} + \nabla \cdot (\rho_f u \otimes u + p\mathbb{I}) = 0, \frac{\partial E}{\partial t} + \nabla \cdot ((E + p)u) = 0, \quad (19)$$

where $\rho_f$ denotes the fluid density, $u$ represents the velocity vector, $p$ is the pressure, and $E$ is the total energy. Viscous effects are disregarded. The boundary conditions are set as follows: $\rho_\infty = 1$, $p_\infty = 1.0$, $M_\infty = 0.8$, and $AoA = 0$, where $M_\infty$ is the Mach number and $AoA$ stands for the angle of attack. A no-penetration condition is applied at the airfoil surface. The airfoil shape is parameterized using the design element method. Specifically, the initial NACA-0012 shape is mapped onto a "cubic" design element with 8 control nodes, and the initial shape is morphed to a different one following the control nodes' displacement field. These control nodes can only move vertically, with displacements following a uniform distribution $d \sim \mathcal{U}[-0.05, 0.05]$. The dataset consists of 1000 training examples and 200 test examples, generated using a second-order implicit finite volume solver. The mesh point locations and the Mach number at these points serve as the input and output data, respectively.

**Hyper-elastic Material (*Elasticity*).** We consider the governing equation of a solid body as:

$$\rho_s \frac{\partial^2 u}{\partial t^2} + \nabla \cdot \sigma = 0$$

where $\rho_s$ is the mass density, $u$ represents the displacement vector, and $\sigma$ is the stress tensor. To close the system, a constitutive model links the strain tensor $\varepsilon$ to the stress tensor. Our study investigates a unit cell problem $\Omega = [0, 1] \times [0, 1]$ with a void at the center. The void's radius follows $r = 0.2 + \frac{0.2}{1 + \exp(\tilde{r})}$ with $\tilde{r} \sim \mathcal{N}(0, 42(-\nabla + 32)^{-1})$. The bottom edge of the unit cell is clamped, and a tension traction $t = [0, 100]$ is applied on the top edge. We use the Rivlin-Saunders material, with energy density parameters $C_1 = 1.863 \times 10^5$ and $C_2 = 9.79 \times 10^3$. Data was generated using a finite element solver with around 100 quadratic quadrilateral elements. The input data, represented as point clouds containing roughly 1000 points, was used to predict stress as the target output.

**Navier-Stokes Equation (*Pipe*).** This scenario involves simulating incompressible flow through a pipe, governed by the Navier-Stokes equations:

$$\frac{\partial v}{\partial t} + (v \cdot \nabla)v = -\nabla p + \mu \nabla^2 v, \quad \nabla \cdot v = 0$$

where $v$ denotes the velocity vector, $p$ is the pressure, and $\mu = 0.005$ represents the viscosity. The inlet imposes a parabolic velocity profile with a maximum velocity of $v = [1, 0]$. A free boundary condition is applied at the outlet, and no-slip conditions are enforced on the pipe's surface. The pipe has a length of 10 and a width of 1. Its centerline is parameterized by 4 piecewise cubic polynomials, controlled by the vertical positions and slopes at 5 spatially uniform nodes. The vertical positions at these control nodes follow a uniform distribution $d \sim \mathcal{U}[-2, 2]$, while the slopes adhere to $d \sim \mathcal{U}[-1, 1]$. The dataset includes 1000 training examples and 200 test examples, generated using an implicit finite element solver with approximately 4000 Taylor-Hood Q2-Q1 mixed elements. The input data comprises the mesh point locations ($129 \times 129$), and the horizontal velocity at these points is the output.

## B EXPERIMENTS SETUPS

To train our model, we follow the experimental setup and hyperparameter choices as outlined in the original CORAL (Serrano et al., 2023) paper. The additional hyperparameters for MARBLE are listed in Table 6.

Table 6: The choices of hyperparameters for GridMix and spatial domain augmentation (SDA)

|  | Hyper-parameter | *Navier-Stokes* | *Shallow-Water* | *NACA-Euler* | *Elasticity* | *Pipe* |
|---|---|---|---|---|---|---|
| GridMix | grid resolution | 8×8 | 8×16 | 32×8 | 8×8 | 8×8 |
|  | basis functions count | 32 | 32 | 64 | 64 | 64 |
|  | latent dimension | 32 | 256 | 128 | 128 | 128 |
|  | basis learning rate | 1e-2 | 1e-2 | 1e-2 | 1e-2 | 1e-2 |
| SDA | sampling ratio | 0.4 | 0.4 | - | - | - |

## C ADDITIONAL VISUALIZATIONS

We present visualizations of the predictions made by MARBLE on the *Navier-Stokes* dataset in Figure 5 and on the *Elasticity* dataset in Figure 6.

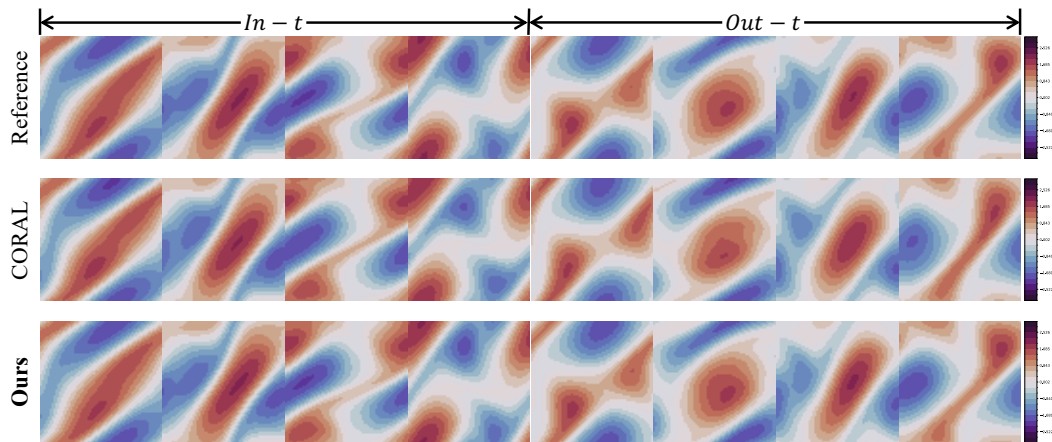

Figure 5: Visualization of predictions of MARBLE on *Navier-Stokes* dataset with $\pi = 20\%$.

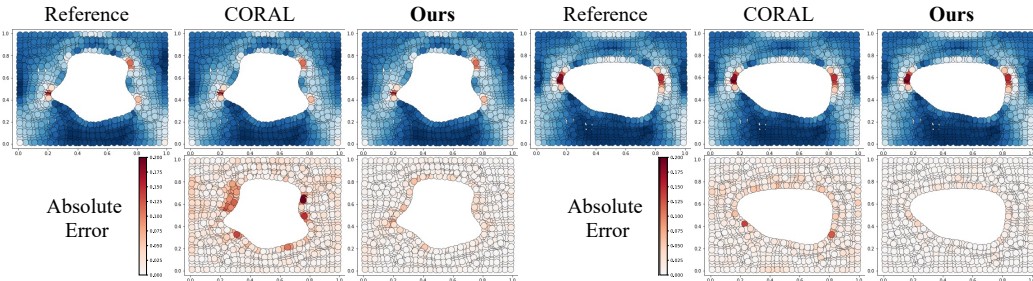

Figure 6: Visualization of predictions of MARBLE on *Elasticity* dataset.

## D EXTENDED RELATED WORK

**Classical Grid-based Methods,** such as finite difference (FDM) (Grossmann, 2007), finite element (FEM) (Huebner et al., 2001), and spectral methods (Shen et al., 2011), are foundational techniques for solving partial differential equations. These approaches discretize the spatial domain into a structured or unstructured grid, allowing the numerical approximation of derivatives and the solution of governing equations. FDM rely on regular grids and approximate derivatives through differences between neighboring grid points. While simple and efficient for structured domains, they

struggle with complex geometries or adaptivity to dynamic domain changes. FEM use unstructured grids and local basis functions to discretize the domain, providing higher flexibility for complex geometries and localized refinement. However, their computational cost can increase significantly with mesh refinement. Spectral Methods decompose the solution into global basis functions (e.g., Fourier or Chebyshev polynomials) and excel in smooth domains due to their high accuracy. Yet, they face challenges in handling discontinuities or irregular domains. Despite their robustness and accuracy, these methods are typically tailored to fixed domains, making generalization to unseen configurations or dynamic problems challenging. Furthermore, their computational complexity scales poorly with increasing grid resolution, especially for high-dimensional problems.

**Spatial Modulation for Neural Fields.** Grid-based representations (Müller et al., 2022) have driven significant progress in reconstructing single continuous signals with neural fields. This foundation has been extended to encode multiple data instances through spatial modulation (Bauer et al., 2023; Kim et al., 2023; Lee et al., 2024a; Wessels et al., 2024). For instance, Bauer et al. (2023) introduced a scalable approach for representing high-resolution images using spatial modulation. Subsequent works, such as Kim et al. (2023) and Lee et al. (2024a), further enhanced the expressive power of this technique by incorporating Transformer-based encoders. Wessels et al. (2024) took this a step further by embedding geometric information into modulation spaces, enabling steerability properties. While the majority of research in this area has focused on computer vision tasks, recent work (Knigge et al., 2024) has explored grounding geometric information within modulation spaces to preserve PDE symmetries, thereby improving generalization and data efficiency for PDE modeling. Our work distinguishes itself by uniquely focusing on enhancing domain generalization while preserving the locality of spatial modulations—an underexplored aspect that is critical for PDE modeling.

## E  EXTENDED COMPARISONS

We compare our method with Factorized-FNO (Tran et al., 2023) in the dynamics modeling setting, as shown in Table 7. Specifically, we use a 12-layer Factorized-FNO with 16 modes and a width of 64. For the irregular grid setting, linear interpolation is used to preprocess the data for compatibility with Factorized-FNO. As observed, Factorized-FNO demonstrates improved performance compared to FNO across all settings. However, our method, MARBLE, consistently achieves superior performance.

Table 7: **Dynamics Modeling** - Test results. Metrics in MSE. The best results are shown in **bold**, while the second-best results are underlined. The "lin. int." abbreviates linear interpolation of irregular grid data onto a regular grid.

| $\mathcal{X}_{tr} \downarrow \mathcal{X}_{te}$ | dataset → | *Navier-Stokes* | |
|---|---|---|---|
| | | *In-t* | *Out-t* |
| | DeepONet | 4.72e-2 ± 2.84e-2 | 9.58e-2 ± 1.83e-2 |
| | FNO | 5.68e-4 ± 7.62e-5 | 8.95e-3 ± 1.50e-3 |
| | Factorized-FNO | 3.18e-4 ± 5.94e-5 | 5.41e-3 ± 1.12e-3 |
| $\pi = 100\%$ | MP-PDE | 4.39e-4 ± 8.78e-5 | 4.46e-3 ± 1.28e-3 |
| regular grid | DINo | 1.27e-3 ± 2.22e-5 | 1.11e-2 ± 2.28e-3 |
| | CORAL | 1.86e-4 ± 1.44e-5 | 1.02e-3 ± 8.62e-5 |
| | MARBLE (Ours) | **3.52e-5 ± 5.31e-6** | **5.04e-4 ± 3.68e-5** |
| | DeepONet | 8.37e-1 ± 2.07e-2 | 7.80e-1 ± 2.36e-2 |
| | FNO + lin. int. | 3.97e-3 ± 8.03e-4 | 9.92e-3 ± 2.36e-3 |
| | Factorized-FNO + lin. int. | 3.42e-3 ± 1.15e-4 | 6.68e-3 ± 2.87e-4 |
| $\pi = 20\%$ | MP-PDE | 3,98e-2 ± 1,69e-2 | 1,31e-1 ± 5,34e-2 |
| irregular grid | DINo | 9.99e-4 ± 6.71e-3 | 8.27e-3 ± 5.61e-3 |
| | CORAL | 2.18e-3 ± 6.88e-4 | 6.67e-3 ± 2.01e-3 |
| | MARBLE (Ours) | **1.62e-4 ± 2.42e-5** | **9.27e-4 ± 1.44e-4** |
| | DeepONet | 7.86e-1 ± 5.48e-2 | 7.48e-1 ± 2.76e-2 |
| | FNO + lin. int. | 3.87e-2 ± 1.44e-2 | 5.19e-2 ± 1.10e-2 |
| | Factorized-FNO + lin. int. | 3.11e-2 ± 3.06e-3 | 4.88e-2 ± 1.58e-3 |
| $\pi = 5\%$ | MP-PDE | 1.92e-1 ± 9.27e-2 | 4.73e-1 ± 2.17e-1 |
| irregular grid | DINo | 8.65e-2 ± 1.16e-2 | 9.36e-2 ± 9.34e-3 |
| | CORAL | 2.44e-2 ± 1.96e-2 | 4.57e-2 ± 1.78e-2 |
| | MARBLE (Ours) | **1.43e-3 ± 5.66e-4** | **4.73e-3 ± 1.33e-3** |

## F    COMPUTATIONAL COMPLEXITY

In this section, we provide an analysis of the computational complexity of our method. Specifically, we unroll a trajectory up to $T = 40$ with a batch size of 1 and report runtime and memory usage in Figure 7. All experiments were conducted on an NVIDIA A100 GPU.

**Time Complexity.**    Our method shows a moderate increase in runtime compared to CORAL. For example, at a resolution of $128 \times 128$, our approach is approximately 1.6× slower than CORAL. This additional overhead arises from steps like spatial modulation and interpolation in our framework. Nonetheless, our method still maintains substantial efficiency compared to traditional numerical methods. The pseudo-spectral method, the numerical baseline used to generate the dataset (Li et al., 2021), requires approximately an order of magnitude more time than our method at the same resolution.

**Memory Usage.**    Our method incurs about 6.5× the memory overhead compared to CORAL across varying resolutions, primarily due to additional parameters and intermediate computations required by our enhanced representation. However, the memory cost scales predictably and remains well within the capabilities of modern GPUs for the tested resolutions, ensuring feasibility for larger datasets and higher resolutions in practical applications. As discussed in Section 5, this memory overhead could potentially be mitigated through techniques such as vector quantization (Takikawa et al., 2022) and tensor factorization (Chen et al., 2022), offering a promising direction for future research.

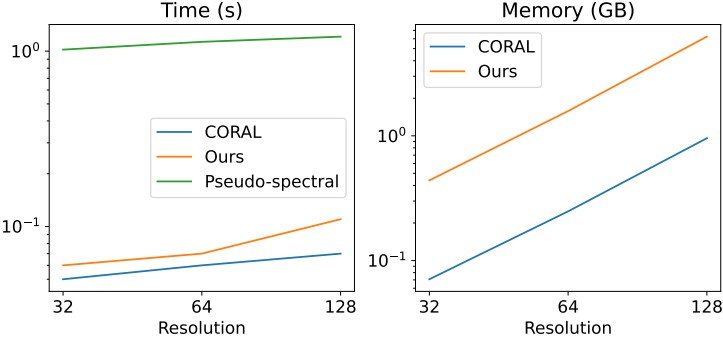

Figure 7: Computational complexity of MARBLE.

## G    VISUALIZATION OF LATENT EMBEDDINGS

To analyze the impact of increasing the number of basis functions on model performance, we visualize the latent embeddings generated with 32 and 64 basis functions in Figure 8. Specifically, we randomly select 5 dimensions from the latent embedding $z$ and plot their changes over time. As observed, the trajectories generated with 64 basis functions exhibit greater fluctuations compared to those with 32 basis functions. This increased variability may complicate the learning process for Neural ODEs, potentially leading to a decrease in forecasting performance.

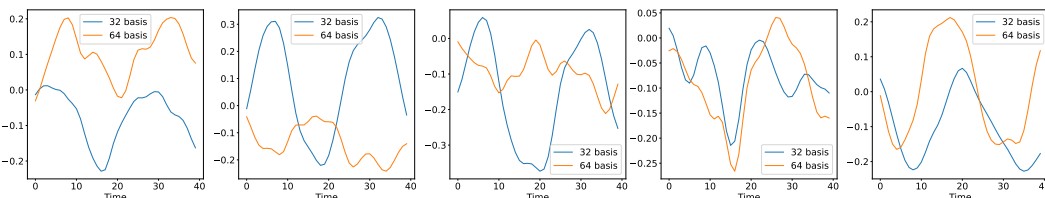

Figure 8: Latent embeddings generated with 32 and 64 basis functions.

