# OpenReview forum: "GridMix: Exploring Spatial Modulation for Neural Fields in PDE Modeling"
_ICLR.cc/2025/Conference — ICLR 2025 Oral_

### Official Review · Reviewer_pmNQ · 2024-10-27

**Soundness:** 3
**Presentation:** 3
**Contribution:** 2
**Rating:** 8
**Confidence:** 4

**Summary:**

In recent years, neural field-based methods for dynamics modelling have seen increasing interesting due to their flexible nature w.r.t. geometry and sparsity of the input. Most methods that have previously been proposed employ conditional neural fields, where a base model is conditioned by a global latent that is shared over all spatial positions in the input function. Although this has shown promising results in complex forecasting settings, the use of global modulation limits the ability of the NeF to reconstruct fine details. As such, authors propose instead learning a spatial basis of modulations in which a sample can be expressed through a combination of coefficients. This allows for location-specific modulation while retaining the globally informed continuous properties of global modulations. Authors show results for a range of experiments, assessing temporal generalization and robustness to subsampling, in which their method consistently improves over baseline architectures.

**Strengths:**

- The method is effective and simple, instead of learning a global modulation which limits fine-grained reconstruction or a purely local latent which limits generalizability, a spatial modulation is expressed in a learned basis. The experimental results show good performance. This approach is general and could be applied to a wide range of other interesting forecasting tasks.
- The paper is well-written and builds towards the method incrementally. The method is very well motivated, also supported by empirical evidence.
- The paper contains illustrative figures that aid in understanding the work.

**Weaknesses:**

- The only limitation I see, although I do not deem it a major one, is the relatively limited novelty of the method. There have been quite a number of previous works exploring spatial modulation for NeF-based reconstruction and downstream performance, notably [1, 2, 3]. For instance, in computer vision NeF literature a widely carried modulation/embedding is hash encoding [4], which your approach to modulation learning seems somewhat related to. Within PDE solving literature, a concurrent work [5] proposes a NeF-based solving framework based on DINo which also learns a spatial latent using MAML, though their approach is more tailored towards PDEs with known symmetries. It might be good to mention these works and compare (maybe in an extended related work section in the appx.), as the (vision) NeF community and the Neural Operator (physics informed learning) communities have been working somewhat in parallel.
- No details are given regarding the computational complexity of the method. Would your method scale easily to higher resolution / larger data? I think this is an important consideration, and as such it would be good to provide details on computational complexity and memory efficiency.

Other than this, I think this work is interesting and shows valuable results for the deep-learning based PDE solving community. I think this paper would be a valuable contribution to the conference.

[1] Bauer, M., Dupont, E., Brock, A., Rosenbaum, D., Schwarz, J. R., & Kim, H. (2023). Spatial functa: Scaling functa to imagenet classification and generation. arXiv preprint arXiv:2302.03130.
[2] Lee, D., Kim, C., Cho, M., & HAN, W. S. (2024). Locality-aware generalizable implicit neural representation. Advances in Neural Information Processing Systems, 36.
[3] Wessels, D. R., Knigge, D. M., Papa, S., Valperga, R., Vadgama, S., Gavves, E., & Bekkers, E. J. (2024). Grounding Continuous Representations in Geometry: Equivariant Neural Fields. arXiv preprint arXiv:2406.05753.
[4] Müller, T., Evans, A., Schied, C., & Keller, A. (2022). Instant neural graphics primitives with a multiresolution hash encoding. ACM transactions on graphics (TOG), 41(4), 1-15.
[5] Knigge, D. M., Wessels, D. R., Valperga, R., Papa, S., Sonke, J. J., Gavves, E., & Bekkers, E. J. (2024). Space-Time Continuous PDE Forecasting using Equivariant Neural Fields. arXiv preprint arXiv:2406.06660.

**Questions:**

- Is there a particular reason you chose to optimize the decoders and neural ode models separately? Would it not be possible to optimize these end-to-end?
- Could you provide details on the computational complexity of your method compared with CORAL? How much overhead is there in terms of memory and time complexity compared to this baseline?

---

> ### Author Response · Authors · 2024-11-20
>
> Thank you for the valuable comments. Our responses to the comments are as follows.
>
> ### Reply to W1:
> We appreciate the insightful comment and thank you for pointing out these relevant works. We agree that spatial modulation has been explored in various contexts. We have carefully considered the suggested references and incorporated a more detailed comparison in the appendix D. To address your concern regarding novelty, **we would like to emphasize that the key distinction of our approach lies in its unique focus on the domain generalization of spatial modulations**. This aspect is under-explored in previous research and is particularly crucial for PDE modeling, where adaptability to diverse domains is essential.
>
> ### Reply to W2 and Q2:
> Thank you for raising this important consideration. We have provided a detailed analysis of computational complexity in Appendix F. Specifically, we unroll a trajectory up to T = 40 with a batch size of 1 and report runtime and memory usage in Figure 7. All experiments were conducted on an NVIDIA A100 GPU.
>
> **Time Complexity**: Our method shows a moderate increase in runtime compared to CORAL. For example, at a resolution of $128 \times 128$, our approach is approximately 1.6× slower than CORAL. This additional overhead arises from steps like spatial modulation and interpolation in our framework. Nonetheless, our method still maintains substantial efficiency compared to traditional numerical methods. The pseudo-spectral method, the numerical baseline used to generate the dataset, requires approximately an order of magnitude more time than our method at the same resolution.
>
> **Memory Usage**: Our method incurs about 6.5× the memory overhead compared to CORAL, primarily due to additional parameters and intermediate computations required by our enhanced representation. However, the memory cost scales predictably and remains well within the capabilities of modern GPUs for the tested resolutions, ensuring feasibility for larger datasets and higher resolutions in practical applications. As discussed in Section 5, this memory overhead could potentially be mitigated through techniques such as vector quantization[3] and tensor factorization[4], offering a promising direction for future research.
>
> ### Reply to Q1
> Thank you for your insightful question. In our work, we chose to optimize the decoder and Neural ODE models separately, following the practices established in prior work[1]. However, exploring end-to-end training is an intriguing direction, and we acknowledge its potential benefits, as demonstrated in [2].
>
> We attempted end-to-end training in our case but encountered challenges. A key issue is that Neural ODE training can be unstable during the early stages due to its auto-regressive nature, which provides less meaningful supervision for the decoder. Additionally, our current training strategies and network architectures were selected specifically for separate training, and adapting them to an end-to-end framework would require careful consideration and redesign.
>
> Since this was not the primary focus of our work, we did not pursue it further but consider it a promising avenue for future exploration.
>
>
> **Reference**
>
> [1] Louis Serrano, Lise Le Boudec, Armand Kassa¨ı Koupa¨ı, Thomas X Wang, Yuan Yin, Jean-No¨el
> Vittaut, and Patrick Gallinari. Operator learning with neural fields: Tackling pdes on general
> geometries. Advances in Neural Information Processing Systems, 36, 2023.
>
> [2] David M Knigge, David R Wessels, Riccardo Valperga, Samuele Papa, Jan-Jakob Sonke, Efstratios
> Gavves, and Erik J Bekkers. Space-time continuous pde forecasting using equivariant neural
> fields. arXiv preprint arXiv:2406.06660, 2024.
>
> [3] Towaki Takikawa, Alex Evans, Jonathan Tremblay, Thomas M¨uller, Morgan McGuire, Alec Ja-
> cobson, and Sanja Fidler. Variable bitrate neural fields. In ACM SIGGRAPH 2022 Conference
> Proceedings, pp. 1–9, 2022.
>
> [4] Anpei Chen, Zexiang Xu, Andreas Geiger, Jingyi Yu, and Hao Su. Tensorf: Tensorial radiance
> fields. In European conference on computer vision, pp. 333–350. Springer, 2022.

---

> > ### Comment · Reviewer_pmNQ · 2024-11-23
> >
> > Thanks for the thorough response, I feel the minor concerns I had were sufficiently addressed. Increased computational efficiency seems a valuable direction of future research extending this work, but it seems you have some idea's on this. I've raised my recommendation. Thanks for the interesting work!

---

### Official Review · Reviewer_oXC3 · 2024-11-02

**Soundness:** 3
**Presentation:** 3
**Contribution:** 3
**Rating:** 8
**Confidence:** 4

**Summary:**

Summary: The paper introduces MARBLE, a new training framework for INR-based neural solvers for PDEs. MARBLE has two components: 1. GridMix and 2. Spatial domain augmentation (SDA). The paper builds off of CORAL [1], primarily changing the shift modulation.

GridMix replaces $\phi_{a} = h_{a}(z_{a_i})$ with a linear combination of learned basis functions. As such, $\phi$ takes the form $\phi_{a}(x) = \Sigma_{m=1}^M c_i^m(z_a)\Phi^m_i(x)$. The set of coefficients $c^m$ are a hypernetwork function of the latent embedding $z_a$ and $\Phi$ functions are learned along with the INR weights. Points on the domain $x$, are binned into patches of $H\times W$, and $\Phi$ is a tensor of shape $H\times W \times C$, representing channel-wise modulation binned according to $H \times W$.

SDA is a form of data augmentation which stochastically drops points from the input domain.

*Experiments*: The paper conducts two sets of experiments: one on learning dynamics (NS, Shallow water) and the second on problems mapping from reference grid to steady states (Euler, NS Pipe, Hyper-elastic material). Ablations are performed over NS.

**Strengths:**

- The problem is of strong interest and the proposed solution is novel.
- Multiple complex datasets are used for experimental results, representing a variety of tasks. The proposed method has substantive improvements over the baselines.
- The ablations indicate a clear improvement to CORAL on both the in and out of distribution time-horizons.

**Weaknesses:**

High level weaknesses, articulated further under questions.
- Some of the mathematical notation and setup for GridMix is confusing.
    - The computation of the position-dependent shift modulation scalar is not clear. The text says: “we derive the position-dependent shift modulation scalar $\phi_i(x)$ at location x through interpolation between neighboring features.” From fig 2, it seems that position x is mapped to a bin which then selects the corresponding modulation value. Please provide a step-by-step explanation as to how x is mapped to the modulation variable, including the details of how the interpolation is performed.
    - Equation 7 is presents $\phi_i(x)$ as independent of $z_{a}$. Equation 7 also omits the hypernetwork, obfuscating the dependence on $z_{a}$. Please clarify in the text that this dependence by providing a supplemental equation that illustrates this relationship or including the hypernetwork in equation 7.
- The baseline models are not consistent between the two classes of experiments.
- Some experimental details are not clear.

**Questions:**

- Why are the models used for the two evaluation settings different? While FNO is a good neural operator baseline, there have been many advancements (e.g., FFNO [2]) which could be run on the dynamics modeling case. *Requested Exp*: Use a better FNO variant for a more apt comparison to spectral neural operators (NOs). Can the authors justify the choice of baselines for each evaluation setting and why certain models were not used in both contexts?
- Table 2: What does “FNO + lin. int.” mean? Additionally, how is an irregular grid being used with FNO? Please add a brief explanation of these terms and methods to either the table caption or main text. A spectral NO designed for irregular meshes would be a more apt baseline (e.g., Geo-FNO [3]). *Requested Exp*: Geo-FNO / FNO variant designed for irregular meshes.
- Table 2: No explanation for the N.A. values.
- “Baseline results for comparison are sourced from Serrano et al. [1]”. What does this mean? Please provide a detailed explanation of how the baseline results were obtained, including any steps to ensure consistency with the original datasets if new data was generated (seems to be indicated in Appendix A.1).
- There are a few cases where the proposed method performs marginally worse than some of the baselines (Shallow water $\pi=5\%$, Pipe). What about these cases makes MARBLE perform worse? It would benefit the paper to include a brief analysis of these specific cases in the discussion section, exploring possible reasons for the underperformance and how future work might add it.
- Dynamics modeling (architecture details): A single modulated INR is used in this case. This is different from Fig 1 and the setup where two modulated INRs are used: $f_{\theta_{a}}$ and $f_{\theta_{u}}$. What is the explicit differences from the general setup in the method section? Please clarify how the input and output functions are handled in this experimental setting.

[1] Serrano, L., Le Boudec, L., Koupaï, A., Wang, T., Yin, Y., Vittaut, J.N., & Gallinari, P. (2024). Operator learning with neural fields: tackling PDEs on general geometries. In Proceedings of the 37th International Conference on Neural Information Processing Systems. Curran Associates Inc..

[2] Alasdair Tran, Alexander Mathews, Lexing Xie, & Cheng Soon Ong (2023). Factorized Fourier Neural Operators. In The Eleventh International Conference on Learning Representations .

[3] Li, Z., Huang, D., Liu, B., & Anandkumar, A. (2024). Fourier neural operator with learned deformations for PDEs on general geometries. J. Mach. Learn. Res., 24(1).

---

> ### Author Response · Authors · 2024-11-20
>
> Thank you for the valuable comments. Our responses to the comments are as follows.
>
> ### Reply to W1:
> Thank you for the comment. We appreciate the opportunity to clarify these points.
> - To derive the modulation parameter $\phi(x)$ for a given spatial location $x$, we first identify the neighboring grid points surrounding this location (in 2D, this involves four points). We then compute $\phi(x)$ by interpolating the values at these neighboring points, using bilinear interpolation in the 2D case.
> - As outlined in Equation 8 of the original submission, we use the latent embedding $z$ to derive the mixture coefficients $c$ through a hypernetwork, which in turn determines the value of $\phi$. For clarity, we have combined Equations 7 and 8 into a single equation.
>
> ### Reply to W2
> Thank you for raising this point. The baseline models differ between the two classes of experiments because each experimental setup evaluates distinct aspects of model performance. We followed prior work[1] to select appropriate baselines for each task.
>
> 1. **In the dynamics modeling setting**, the focus is on assessing temporal forecasting capabilities. For this, we compare our method against:
> - Two classical neural operators (DeepONet and FNO),
> - A well-designed auto-regressive graph-based model (MP-PDE),
> - Two coordinate-based approaches (DINo and CORAL).
>
> 2. **In the geometry-aware inference setting**, the focus shifts to evaluating models' abilities to generalize across varying geometries. Here, we compare with:
> - Two regular-grid baselines (FNO and UNet)
> - Two improved FNO variants, Geo-FNO and Factorized-FNO, which are specifically designed for enhanced performance on geometry-aware tasks.
> - A coordinate-based approach (CORAL)
>
> **The selection of baselines is tailored to align with the primary goal of each experiment**, ensuring fair and meaningful comparisons. Based on your feedback, we have incorporated the following updates:
> 1. Added Factorized-FNO as a baseline in the dynamics modeling setting (see **Reply to Q1**).
> 2. Provided an analysis explaining why Geo-FNO is unsuitable for irregular-grid settings in dynamics modeling (see **Reply to Q2**).
>
> We hope these clarifications address your concerns and enhance the understanding of our baseline choices.
>
> ### Reply to Q1:
> Thank you for the suggestion. We have included a comparison with Factorized-FNO (FFNO) in the dynamics modeling setting, as shown in the following table. Specifically, we use a 12-layer Factorized-FNO with mode=16 and width=64. As observed, Factorized-FNO demonstrates improved performance compared to FNO across all settings. However, our method, MARBLE, consistently achieves superior performance. We have included these results in the Appendix E.
>
> |Setting|Method|N-S In-t|N-S Out-t|
> |:-:|:-:|:-:|:-:|
> |$\pi=100\%$|FNO|5.68e-4 ± 7.62e-5|8.95e-3 ± 1.50e-3|
> | |Factorized-FNO|3.18e-4 ± 5.94e-5 | 5.41e-3 ± 1.12e-3|
> | |MARBLE|**3.52e-5 ± 5.31e-6**|**5.04e-4 ± 3.68e-5**|
> |$\pi=20\%$|FNO+lin. int.|3.97e-3 ± 8.03e-4|9.92e-3 ± 2.36e-3|
> | |Factorized-FNO+lin. int.|3.42e-3 ± 1.15e-4|6.68e-3 ± 2.87e-4|
> | |MARBLE|**1.62e-4 ± 2.42e-5**|**9.27e-4 ± 1.44e-4**|
> |$\pi=5\%$|FNO+lin. int.|3.87e-2 ± 1.44e-2 |5.19e-2 ± 1.10e-2|
> | |Factorized-FNO+lin. int.|3.11e-2 ± 3.06e-4|4.88e-2 ± 1.58e-3|
> | |MARBLE|**1.43e-3 ± 5.66e-4**|**4.73e-3 ± 1.33e-3**|
>
> ### Reply to Q2:
> Thank you for your suggestion. “FNO + lin. int.” refers to preprocessing irregular grid data by linearly interpolating it onto a regular grid, which is then used as input for the standard FNO. To ensure clarity, we have included this explanation in the table caption.
>
> Regarding Geo-FNO, while it is designed to handle irregular meshes, it is not suitable for this setting due to the following reasons:
> 1. In our setup, the irregular points are sampled from a regular grid. Transforming these irregular points back into a regular grid, as required by Geo-FNO’s deformation mechanism, disrupts their original structure.
> 2. In our setup, the same irregular grid is used across all training samples, while a different irregular grid is employed for testing samples. Geo-FNO relies on learnable grid deformation, which is prone to overfitting to the training grid due to the limited diversity of training grids in our case. Consequently, it performs poorly when generalizing to unseen testing grids.
>
> Given these limitations, we sought a more robust alternative. To this end, we have incorporated Factorized-FNO as an enhanced baseline as shown in **Reply to Q1**. Linear interpolation is used to preprocess the irregular grid for compatibility with this approach.  This approach ensures a fair and meaningful comparison while maintaining compatibility with our irregular grid setting

---

> ### Author Response · Authors · 2024-11-20
>
> ### Reply to Q3:
> Thank you for the comment. We follow the practice in prior work, CORAL[1], where the same entries are reported as "N.A." The "N.A." values in Table 2 indicate that the corresponding results are "Not Available." Specifically, for the $\textit{Shallow-Water}$ dataset, the data is represented in a 2D spherical grid (latitude and longitude), which is not directly compatible with the linear interpolation method we used, resulting in the "N.A." values.
>
> We hope this clarifies the reason behind the missing entries. Please let us know if further details are needed.
>
> ### Reply to Q4:
> Thank you for your question. In our work, we adhered to the experimental setup and utilized the original datasets provided by Serrano et al.[1] to maintain consistency in our comparisons. We reproduced their experiments by following the training recipes for CORAL and other baselines as detailed in their Appendix B. Our reproduction closely aligned with the reported results in their paper. Consequently, we directly used their reported results as baselines for our comparison. No additional data was generated for these experiments. For clarity and reproducibility, we include detailed dataset preparation steps in Appendix A.1, so our approach remains fully self-contained.
>
> ### Reply to Q5:
> Thank you for highlighting this point. We acknowledge that in certain settings, such as Shallow Water ($\pi=5\%$) and Pipe, MARBLE’s performance was slightly below some baselines. Here is our analysis:
> 1. Shallow Water ($\pi=5\%$): MARBLE underperforms DINo in the In-t setting but outperforms it in the Out-t setting. This difference primarily stems from their distinct approaches to embedding optimization in the auto-decoder. In DINo, each sample’s embedding is initialized once and optimized throughout training without reset, resulting in modulation space that are finely tuned to the training data. While this strategy can yield highly optimized embeddings, it also increases the risk of overfitting, particularly when the input data is sparse, limiting temporal extrapolation. In contrast, MARBLE follows a meta-learning strategy as outlined in Section 3.2, where each inner-loop re-initializes embeddings from scratch and optimizes them for only three steps. This method, combined with an outer-loop that optimizes the INR weights conditioned on the inner-loop embeddings, promotes smoothness in the modulation space and enhances temporal extrapolation, leading to stronger performance in the Out-t setting.
> 2. Pipe Dataset: MARBLE performs below GEO-FNO and Factorized-FNO on the Pipe dataset, likely due to limitations of the SIREN backbone used. The Pipe dataset exhibits high frequencies predominantly along the vertical dimension. While SIREN is effective for isotropic frequency distributions, it may struggle with data featuring strong directional anisotropy due to its sensitivity to frequency-related hyperparameters. To address this, future work could explore improved INR designs[2] capable of capturing directional frequency variations.
>
> We have incorporated these analyses into the experiments and discussion sections to provide further clarity. Thank you again for your valuable feedback.
>
> ### Reply to Q6:
> Thank you for your insightful question. In this specific experiment, our goal is to predict the future states (output function) of a physical quantity, given its current state (input function). The data we utilize consists of various initial conditions and time instances, all sampled from a similar underlying distribution. As a result, we can assume that the **input and output functions come from the same function space**.
>
> Given the **relatively homogeneous nature of this dataset**, we found that a single modulated INR was sufficient to capture the underlying variations. This simplified architecture allows for efficient training, while still achieving strong performance.
>
> In contrast, the general setup outlined in our method section, involving two modulated INRs, is designed for more complex scenarios where **the input and output functions may exhibit distinct characteristics**. This approach provides greater flexibility and can be applied to a wider range of tasks.
>
> **Reference**
>
> [1] Louis Serrano, Lise Le Boudec, Armand Kassa¨ı Koupa¨ı, Thomas X Wang, Yuan Yin, Jean-No¨el
> Vittaut, and Patrick Gallinari. Operator learning with neural fields: Tackling pdes on general
> geometries. Advances in Neural Information Processing Systems, 36, 2023.
>
> [2] Junwoo Cho, Seungtae Nam, Hyunmo Yang, Seok-Bae Yun, Youngjoon Hong, and Eunbyung Park.
> Separable physics-informed neural networks. Advances in Neural Information Processing Sys-
> tems, 36, 2024.

---

> ### Comment · Reviewer_oXC3 · 2024-11-20
> **Reply to the authors**
>
> I thank the authors for the detailed responses to my questions and for updating the manuscript. I have raised my score from a 6 to an 8.

---

### Official Review · Reviewer_M2jE · 2024-11-03

**Soundness:** 4
**Presentation:** 4
**Contribution:** 3
**Rating:** 8
**Confidence:** 3

**Summary:**

To address the limitations of global modulation, this paper proposes GridMix technology, a novel spatial modulation method. GridMix combines the advantages of grid-based representation and spatial modulation by regularizing the modulation space into a set of shared basis functions, thereby enhancing the model's ability to learn global structures. Additionally, a spatial domain enhancement strategy is introduced to simulate domain changes that may occur during inference, further improving the model's generalization capability. Experiments demonstrate that GridMix excels in dynamic modeling and geometric prediction tasks, particularly in cases of temporal extrapolation and spatial subsampling, outperforming existing models.

**Strengths:**

1. Innovation: The introduction of GridMix overcomes the shortcomings of traditional global modulation in capturing local features by incorporating grid-based representation.
2. Experimental Validation: The effectiveness and versatility of MARBLE are validated through extensive experiments, including dynamic modeling and geometric prediction. The experimental design is robust, covering various testing scenarios and enhancing the credibility of the results.
3. Algorithm Flexibility and Universality: The MARBLE framework demonstrates applicability across multiple tasks by combining GridMix and spatial domain enhancement, indicating its broad potential for application in PDE modeling.

**Weaknesses:**

1. Insufficient Literature Review: There is a lack of introduction to grid-based methods, such as discretization techniques and computational efficiency.
2. Complexity of Proposed Method: Although the proposed new method is innovative, its complexity may lead to challenges in implementation, particularly regarding parameter tuning and optimization in different application scenarios. It is unclear whether this method retains a speed advantage compared to numerical simulations.

**Questions:**

1. What are the basic settings of the grid? The current article only mentions grid density, but it lacks details on discretization methods, grid shapes, and adaptive methods (how to handle complex geometries). Is the discretization method used by the authors consistent with those in numerical simulations?
2. What is the computational efficiency of the proposed method? There is a need to compare the proposed method with numerical simulations and other data-driven PDE modeling methods.

---

> ### Author Response · Authors · 2024-11-20
>
> Thank you for the valuable comments. Our responses to the comments are as follows.
>
> ### Reply to W1:
> Thank you for your comment. We have included a detailed introduction to classical grid-based methods, including finite difference, finite element and spectral methods, in Appendix D for further context and elaboration. Please let us know if there are any specific aspects you would like us to expand upon further.
>
>
> ### Reply to W2 and Q2:
> Thank you for highlighting these important points. We have provided a detailed analysis of computational complexity in Appendix F. Specifically, we evaluated the runtime of our method by unrolling a trajectory up to  T = 40  with a batch size of 1, as shown in Figure 7. All experiments were conducted on an NVIDIA A100 GPU.
>
> Our method exhibits a moderate increase in runtime compared to CORAL. For instance, at a resolution of  $128 \times 128$, it is approximately 1.6× slower than CORAL. This additional computational cost stems from steps such as spatial modulation and interpolation within our framework. Despite this overhead, our method still maintains substantial efficiency compared to traditional numerical methods. For example, the pseudo-spectral method, the numerical baseline used to generate the dataset, requires roughly an order of magnitude more time than our method at the same resolution.
>
> ### Reply to Q1:
> Thank you for your question regarding the grid settings.
> 1. **Grid Shape and Density**: The grid we employ is a regularly spaced, uniform grid, consistent across all experiments. The density of this grid refers to the level of resolution (H*W for the 2D case), which determines how finely we sample and represent the spatial field. The choice of uniform density aims to balance computational efficiency with the desired detail level.
> 2. **Discretization Method**: Since our approach does not involve traditional PDE solving, we do not use explicit discretization techniques such as finite difference or finite element methods. Instead, the grid is uniformly dense to provide consistent feature sampling, which helps in representing spatial variations for modulated INRs. Each cell in this grid contains a learnable feature, allowing us to implicitly encode spatial information that the model can leverage for scene reconstruction.
> 3. **Handling Complex Geometries**:  Traditional simulation methods often employ adaptive grids to refine mesh elements around complex geometries and capture fine-grained details. In contrast, in the Geometry-Aware Inference section, we treat each geometry as a deformation function $X_i$, which describes how the geometry is obtained by transforming a shared, uniform reference domain $X$. We model the family of geometry deformation functions using a modulated INR, where $X_i(x) = f_{\theta, \phi_i}(x)$ with $x \in X$. By learning the deformation function $f_{\theta, \phi_i}$ on the uniform grid of the reference domain, we can implicitly represent complex geometric variations. Each geometry $X_i$ is then associated with a grid-based modulation parameter $\phi_i$, which is defined in the reference domain and takes the form of a uniform grid. This approach enables us to efficiently handle diverse geometries without the need for explicit grid adaptation.

---

> > ### Comment · Reviewer_M2jE · 2024-11-26
> >
> > Thanks for your answer, which address my concerns. I am maintaining my current score.

---

### Official Review · Reviewer_bgUL · 2024-11-09

**Soundness:** 3
**Presentation:** 3
**Contribution:** 3
**Rating:** 8
**Confidence:** 2

**Summary:**

The paper proposes an new architecture called GridMix, which enhances implicit neural fields (INRs) for solving PDEs. Building on multi-resolution INR advances from visual computing ([1], [2]), the authors adapt these methods to improve PDE modeling. More specifically, they solve the problem using CORAL[3] approach, where each of the INR is encoded using a multi-resolution feature grid (for spatial modulation), combined with a new spatial domain augmentation strategy to boost generalization across spatial domains. Experiments demonstrate GridMix’s efficacy in dynamic modeling tasks (temporal extrapolation and spatial interpolation) and in steady-state predictions based on domain geometry.

**Strengths:**

– Adaptation of multi-feature grids for auto-decoding: the paper presents a novel approach for leveraging multi-feature grids in the auto-decoding setting, effectively adapting INRs to capture complex spatial variations while maintaining efficiency.
– Extensive comparisons and ablations study
– Modeling spatial modulation using basis functions (and their coefficients) looks promising

**Weaknesses:**

– Some quantitative results, especially those comparing single-resolution grids with global modulation in Table 1, appear counterintuitive. The single-channel GridMix approach yields lower performance in some cases compared to simpler global modulation, which contradicts the initial motivation that spatial features should perform better.
– Missing mentioning Factor Fields [4] paper that also works in similar framework and solves similar problems but not in PDE domain

**Questions:**

– In Table 1, why single resolution feature grid (Single-channel GridMix) gives worse results
than Global Modulation? As the spatial feature representation should give already good results
given the initial motivation
– I dont fully understand the explanation of why more basis functions lead to worse results at
some point? As in case there are too many basis, they can be zeroed out with corresponding
weights
- Figure 3 is hard to read – neet legend on the Y axis. It is not clear to me from the footnote what’s going
on.


References:
[1] “Neural Geometric Level of Detail: Real-time Rendering with Implicit 3D Shapes” by Towaki
Takikawa et. al.
[2] “Instant Neural Graphics Primitives with a Multiresolution Hash Encoding” by Thomas Muller
et. al.
[3] “Operator Learning with Neural Fields: Tackling PDEs on General Geometries” by Louis
Serran et. al.
[4] “Factor Fields: A Unified Framework for Neural Fields and Beyond” by Anpei Chen et. al.

---

> ### Author Response · Authors · 2024-11-20
>
> Thank you for the valuable comments. Our responses to the comments are as follows.
>
> > W1: Some quantitative results, especially those comparing single-resolution grids with global modulation in Table 1, appear counterintuitive. The single-channel GridMix approach yields lower performance in some cases compared to simpler global modulation, which contradicts the initial motivation that spatial features should perform better.
>
> > Q1: In Table 1, why single resolution feature grid (Single-channel GridMix) gives worse results than Global Modulation? As the spatial feature representation should give already good results given the initial motivation
> ### Reply to W1 and Q1:
> Thank you for this question. **Single-channel GridMix can be viewed as a regularized version of vanilla spatial modulation, where the modulation space is deliberately constrained to enhance domain generalization**. This limited modulation space, however, imposes constraints on its performance. As shown in Table 1, the vanilla spatial modulation indeed achieves better reconstruction in the training domain  $\mathcal{X}_{tr}$ , with an error of  3.95e-5  compared to  1.32e-4  for Global Modulation. This supports the original motivation for spatial modulation’s use in capturing finer details.
>
> However, vanilla spatial modulation struggles to generalize to unseen spatial domains. To mitigate this, Single-channel GridMix constructs spatial feature representations as a linear combination of shared basis functions across all samples. **This design implicitly regularizes the modulation space, improving generalization to new spatial domains at the expense of reduced representational capacity**. This trade-off ultimately leads to a slightly lower performance on the original domain but increases robustness to the change of spatial domain. To further enhance reconstruction performance while preserving this robustness, we propose Multi-channel GridMix, which demonstrates superior results compared to Global Modulation.
>
> > W2: Missing mentioning Factor Fields [4] paper that also works in similar framework and solves similar problems but not in PDE domain.
> ### Reply to W2:
> Thank you for pointing out the Factor Fields [4] paper. We appreciate the opportunity to address this. While Factor Fields decomposes a signal into components such as coefficient fields and basis functions, **our work performs a similar decomposition within the modulation space rather than the signal space**. Additionally, Factor Fields primarily addresses tasks outside the PDE domain, whereas our approach **focuses on PDE-related challenges, with an emphasis on the domain generalization of spatial modulations**.
>
> We have included a comparison with Factor Fields in the related work section to highlight the differences in application focus and the unique contributions of our method. Please feel free to let us know if further clarifications or expanded analyses would be helpful.
>
> > Q2: I dont fully understand the explanation of why more basis functions lead to worse results at some point? As in case there are too many basis, they can be zeroed out with corresponding weights.
> ### Reply to Q2:
> Thank you for your question. A possible reason why adding more basis functions leads to worse results at some point is the **increased complexity of the latent trajectories**. While it’s theoretically possible to zero out redundant basis functions through corresponding weights, in practice, the model often utilizes this additional capacity to refine the reconstruction, resulting in more complex latent trajectories. **These trajectories can become overly intricate, potentially making them difficult for the NeuralODE to learn effectively**. As shown in Appendix G, the trajectories generated with 64 basis functions fluctuate more than those with 32, which may hinder forecasting performance.
>
> > Q3: Figure 3 is hard to read – need legend on the Y axis. It is not clear to me from the footnote what’s going on.
> ### Reply to Q3:
> Thank you for your feedback. We’ve refined Figure 3 to improve its readability and have added a legend on the Y-axis to clarify the data presented. Additionally, we revised the footnote for greater clarity on the figure’s context and content.
>
> To summarize, Figure 3 illustrates key findings from Table 1:
> 1. Spatial modulation significantly reduces reconstruction error on  $\mathcal{X}\_{tr}$  but struggles to capture global information and generalize to  $\mathcal{X}_{te}$.
> 2. GridMix mitigates the overfitting issue associated with spatial modulation while achieving superior performance compared to global modulation.
>
> Please let us know if there are any further improvements or clarifications we can provide.

---

> > ### Comment · Reviewer_bgUL · 2024-11-25
> >
> > I thank the authors for addressing my concerns. I have raised my score from 6 to 8.

---

### Meta-Review · Area_Chair_dFSS · 2024-12-17

**Metareview:**

A common use of neural fields for PDE modeling is to represent functions using global modulations (a.k.a. conditioning in visualization/graphics/vision applications) and learn a map between modulations. This work focuses on the first part and suggests replacing global modulations with spatial ones. This is motivated by advances in neural representations applications in vision and graphics where spatial conditioning is used to increase fidelity and approximation power at different levels of details. In the experimental part the paper demonstrates the usefulness of the suggested method in dynamic modeling and geometric prediction.

The reviewers overall appreciated the novel adaptation of spatial conditioning and the extensive experimental setup demonstrating the efficacy of the suggested method. One comment is that the novelty of the contribution of the method is somewhat limited due to the rather common usage of spatial conditioning in neural fields.

**Additional Comments On Reviewer Discussion:**

No additional comments.

---

### Decision · Program_Chairs · 2025-01-22

Accept (Oral)